

# Timescale dependence of environmental controls on methane efflux in Poyang Lake, China

**Lixiang Liu[1], Ming Xu[1,2*], Rui Shao[1, 3]**

[1]Key Laboratory of Ecosystem Network Observation and Modeling, Institute of

Geographic Sciences and Natural Resources Research, Chinese Academy of Sciences,

Beijing 100101, China

[2]Center for Remote Sensing and Spatial Analysis, Department of Ecology, Evolution

and Natural Resources, Rutgers University, New Brunswick, NJ 08901, USA

[3]University of Chinese Academy of Sciences, Beijing 100049, China

**Correspondence to: Ming Xu (mingxu@ igsnrr.ac.cn)**



**Abstract**

Lakes are an important natural source of $CH_4$ to the atmosphere. However, the long-term $CH_4$ efflux in lakes has been rarely studied. In this study, the $CH_4$ efflux in Poyang Lake, the largest freshwater lake in China, was measured continuously over a 4-year period by using the floating chamber technique. The mean annual $CH_4$ efflux

throughout the 4 years was 0.54 mmol $m^{-2}$ $day^{-1}$, ranging from 0.47 to 0.60 mmol $m^{-2}$ $day^{-1}$. The $CH_4$ efflux had a high seasonal variation with an average summer (June to August) efflux of 1.34 mmol $m^{-2}$ $day^{-1}$ and winter (December to February) efflux of merely 0.18 mmol $m^{-2}$ $day^{-1}$. The efflux showed no apparent diel pattern, although most of the peak effluxes appeared in the late morning, from 10:00 h to 12:00 h.

Multivariate stepwise regression on a seasonal scale showed that environmental factors, such as sediment temperature, sediment total nitrogen content, dissolved oxygen, and total phosphorus content in the water, mainly regulated the $CH_4$ efflux. However, the $CH_4$ efflux only showed a strong positive linear correlation with wind speed within a day on a bihourly scale in the multivariate regression analyses but

almost no correlation with wind speed on diurnal and seasonal scales.

**Keywords:** Methane, Sediment temperature, Temperature sensitivity, Substrate availability, Wind speed

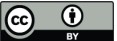



## 1. Introduction

Methane ($CH_4$) contributes to about 20% of global warming in terms of radiative

forcing, and its concentration in the atmosphere increased at a rate of 0.5 ppb year$^{-1}$ in

1999–2006; this rate rapidly increased to 6 ppb year$^{-1}$ from 2007 to 2011 (IPCC,

2013). Although the total global lake area accounts for approximately 3.7% of the

Earth's nonglaciated land area (Verpoorter et al., 2014), $CH_4$ emissions from global

lakes account for up to 14.9% of natural $CH_4$ emissions (IPCC, 2013). However, this

estimate has been associated with large uncertainties because of the high spatial and

temporal variations of $CH_4$ emissions and the insufficient long-term measurements of

$CH_4$ effluxes, especially in tropical and subtropical lakes (Yang et al., 2011; Ortiz–

Llorente and Alvarez–Cobelas, 2012; Bastviken et al., 2015; Li and Bush, 2015).

$CH_4$ effluxes in lakes feature high temporal variations (K äki, 2001; Xing et al.,

2004; Duan et al., 2005; Xing et al., 2005, 2006; Palma–Silva et al., 2013). For

example, previous studies found that the minimum and maximum $CH_4$ effluxes over a

day were −1.36 and 128.85 mmol m$^{-2}$ day$^{-1}$, respectively (Xing et al., 2004; Duan et

al., 2005; Chen et al., 2007; Podgrajsek et al., 2014a, 2014b); even larger variations

were found on a seasonal scale (Xing et al., 2005, 2006; Duan et al., 2005; Ortiz–

Llorente and Alvarez–Cobelas, 2012; Wik et al., 2014). These large variations in $CH_4$

effluxes highlight the importance of frequent and long-term measurements (Bastviken

et al., 2008; Chen et al., 2013; Bastviken et al., 2015). Unfortunately, most earlier

studies on $CH_4$ emissions were based on short-term measurements, ranging from daily

to seasonal scales, and were conducted during the day time (Xing et al., 2004; Duan et

al., 2005; Xing et al., 2005; Schrier–Uijl et al., 2011; R õõm et al., 2014). To our

knowledge, multi-year measurements of $CH_4$ effluxes have only been conducted in

high-latitude lakes, and few studies on tropical and subtropical lakes, especially large



ones, had measurement durations longer than one year.

The magnitude of $CH_4$ emission mainly depends on the dynamic balance between the microbial processes of $CH_4$ production, oxidation, physical transportation from the anaerobic zone to the atmosphere in lakes, and regulation by multiple, interconnected physical, chemical, and biological variables (Sun et al., 2012; Liu et al., 2013; Serrano–Silva et al., 2014; Rasilo et al., 2015). $CH_4$ production and oxidation

are microbial processes regulated by organic carbon loading, dissolved organic matter, lake nutrient status, and N availability (Bridgham et al., 2013; Liu et al., 2013; Hershey et al., 2014; Rasilo et al., 2015); temperature (Liikanen et al., 2003; Marotta et al., 2014; Yvon–Durocher et al., 2014); lake depth and size (Juutinen et al., 2009; Rasilo et al., 2015); pH, $O_2$, $NO_3^{2-}$, $Fe^{3+}$, and $SO_4^{2-}$ in the sediment and water column

(van Bodegom and Scholten 2001; Schrier–Uijl et al., 2011; Bridgham et al., 2013); and populations and potential activities of methanogens and methanotrophs (Segers, 1998; van Bodegom and Scholten, 2001; Liu et al., 2015, 2016). $CH_4$ transportation is driven by three major mechanisms, namely, molecular diffusion, bubble ebullition, and plant-mediated transportation (Bridgham et al., 2013; Chen et al., 2013; Zhu et al.,

2016). These mechanisms are affected by water stratification and seasonal overturns of the water mass, which are determined by temperature, wind-forced mixing, water depth, boundary layer dynamics, hydrostatic pressure, and different vascular plants (Juutinen et al., 2009; Zhu et al., 2016). Most studies examined $CH_4$ emissions and their influencing factors in small lakes because of their large contribution to the global

$CH_4$ budget (Bastviken et al., 2004; Downing et al., 2010; Bartosiewics et al., 2015; Holgerson et al., 2016). However, few studies reported temporal $CH_4$ emissions and their key regulating factors at different temporal scales in large lakes. Therefore, investigating the impacts of physical and biological factors on temporal $CH_4$ effluxes



based on long-term measurements in a large lake is also important to estimate lake

$CH_4$ emissions.

Poyang Lake, a subtropical lake, is the largest freshwater lake in China, but its

annual $CH_4$ emissions have not been adequately measured. In this study, we measured

the $CH_4$ efflux over the course of 4 years in Poyang Lake to (1) examine the annual

$CH_4$ efflux; (2) explore the $CH_4$ efflux dynamics, including diel, seasonal, and

inter-annual variations; and (3) quantify the relationships between the $CH_4$ efflux and

environmental factors, and identify the possible factors driving $CH_4$ effluxes at

different temporal scales.






## 2. Materials and methods

2.1. Site description

Poyang Lake (28°22′–29°45′N, 115°47′–116°45′E) is located in Southern China
in Jiangxi Province, with a surface area of 3283 km$^2$ and a total catchment area of
162,000 km$^2$, which is separated to the northern and southern parts by the Songmen
Mountain. Poyang Lake receives water input from five main tributaries, namely, the
Raohe River, Xinjiang River, Fuhe River, Ganjiang River, and Xiushui River. The
climate is humid subtropical with a mean annual temperature of 17.5 ℃ and an
annual precipitation of 1680 mm (Ye et al., 2011). Vegetation in the lake is composed
of macrophytes, including *Carex* sp. and *Artemisia selengensis* in the hydrophyte zone,
and the main submerged aquatic macrophytes, including *Ceratophyllum demersum*,
*Potamogeton malaianus*, *Potamogeton crispus*, and *Hydrilla verticillata* (Wang et al.,
2011).

This study was conducted near the Poyang Lake Laboratory of the Wetland
Ecosystem Research Station (operated by the Chinese Academy of Sciences), which is
located in the northern sub-basin of Poyang Lake in Xingzi County, Jiangxi Province
(Fig. 1). The five tributaries flow into the lake in the southeast of Xingzi County,
which then joins with the Yangtze River. The water level fluctuated dramatically from
7.78 m to 18.57 m above sea level (Wu Song) between the wet (April to September)
and dry seasons (October to March) during the study period because of rainfall and
Three Gorge management. Poyang Lake is not stratified (Zhu and Zhang, 1997), with
mean and maximum depths of 8 and 23 m, respectively. The concentrations of total
nitrogen (TN), total phosphorous (TP), suspended solids (SS), and chlorophyll *a* (Chl
*a*) in the lake were 3.45, 0.11, 39.98, and 9.04 mg L$^{-1}$, respectively (Yao et al., 2015).

2.2. CH$_4$ efflux measurements





The $CH_4$ efflux was measured using floating chambers, including both ebullition

and diffusive fluxes (Bastviken et al., 2004, 2010). The floating chamber was

fabricated using a PVC pipe 100 cm in length and 20 cm in diameter with Styrofoam

floats attached to the sides. The floating chambers were inserted 80 cm into the water

and 20 cm above the water surface to minimize the perturbation of the surface water

flow to the pressure inside the chambers. We tested the chamber system with different

insertion depths in the laboratory and field, and found that the current depth of about

80 cm could effectively prevent the impacts of the surrounding Styrofoam floats while

maintaining the chamber balance in moderate winds. A similar design of floating

chambers was used in previous studies (Lorke et al., 2015; Zhao et al., 2015). Zhao et

al. (2015) have recently conducted a systematic comparison of the effects of chamber

shape, dimension, and insertion depth into the water on $CH_4$ effluxes and found that

insertion depth only slightly affects the $CH_4$ efflux measured in the Three Gorges

Reservoir when wind speed is relatively low. In the current study, the insertion depth

was deeper than those of previous studies to avoid the impact of waves in Poyang

Lake on the chamber body. Earlier studies also found that floating chambers should be

seated at the water surface with minimal insertion into the water in a flowing-water

system to minimize the "drag" effect of flowing water on chamber pressure

(Bastviken et al., 2010; Vachon et al., 2013; McGinnis et al., 2015). Except for some

waves, the water in Poyang Lake did not have an apparent directional flow during the

measurement period. A detailed description of the floating chamber system can be

found in Liu et al. (2013).

We collected a gas sample (ambient concentration) immediately after the

chamber was closed and three other samples at a 20 min interval for 1 h. The gas was

extracted into a 12 mL evacuated glass vial by a 2 mL syringe needle with an air



pump, which enhanced the pressure in the vial to 3 bars. Subsequently, the air samples

were transported immediately to a laboratory for $CH_4$ concentration analysis. The $CH_4$

concentration was measured using a gas chromatograph equipped with a flame

ionization detector (GC7890A, Agilent Technologies, Inc., Santa Clara, CA, USA).

We used nitrogen ($N_2$) as the carrier gas, which ran at a flow rate of 30 mL min$^{-1}$. We

calibrated the gas chromatograph for every four samples with a calibration gas of 2.03

ppm at 99.92% precision (China National Research Center for Certified Reference

Materials, China). The oven and detector temperatures of the GC were set to 55 ℃

and 250 ℃, respectively.

     Calculation of the $CH_4$ efflux was based on the $CH_4$ concentration of the four

samples using a linear regression model. Data quality control was conducted

following the method of Rasilo et al. (2015) before the regression models were fitted.

As a result, most of the models performed satisfactorily, with a coefficient of

determination ($R^2$) greater than 0.95. In case of ebullition, the $CH_4$ concentration

inside the chamber would deviate from the normal trend. Most of the $CH_4$

concentrations measured immediately after the ebullition point slightly decreased

mainly because of the $CH_4$ diffusion back to water when the $CH_4$ concentration inside

the chamber space increased suddenly from bubbling. To include the

ebullition-induced $CH_4$ emissions, we only used two measured concentrations, the

first measurement (ambient concentration) and an ebullition-adjusted concentration, in

calculating the $CH_4$ efflux when ebullition occurred inside the chamber. The

ebullition-adjusted concentration was obtained by adding the diffusion-induced

concentration increment, which is a correction term, to the measured concentration

when ebullition occurred. The total $CH_4$ efflux, which includes both ebullition and

diffusive effluxes, was calculated on the basis of the slope of the concentration change



during the whole period when the chamber was closed (Fig. 2). Specifically, when

ebullition occurred during the first 20 min, we obtained the ebullition-adjusted

concentration by summing up concentration on 20 min and the 2-fold incremental

concentration between the third and fourth sampling times. When the ebullition

occurred at the third sampling, we summed up the concentration at 40 min and the

incremental concentration between the first and second sampling times. When the

ebullition occurred at the fourth sampling, we used the first and fourth sampling

concentrations directly to calculate the slope of the total efflux.

Samplings took place at a monthly interval from January 2011 to December 2014

at three sites in Poyang Lake (Fig. 1): site A (Luoxingdun: 29 °3′29″N, 116 °16′49″E),

site B (Mantianxing: 29 °34′25″N, 116 °13′29″E), and site C (Huoyanshan: 29 °39′0″N,

116 °16′11″E). The mean water depth in our sampling sites was 3 m. The sampling

sites lacked aquatic plants. Our previous study examined the spatial pattern of the $CH_4$

efflux in the lake (Liu et al., 2013). Therefore, we focused on the long-term dynamics

of $CH_4$ efflux in the current study. At each site, four chambers were placed

approximately 10 m away from a small boat to minimize disturbance. Measurements

were conducted from early morning to late afternoon with about 6 cycles of

measurements for each chamber, except for days when the diel-cycle measurements

were taken. We conducted four 24 h measurements at the three sites in 24–25 July

2011, 5-6 September 2012, 13–14 January 2013, and 14–15 January 2015 to examine

the diel variations of $CH_4$ effluxes. These measurements were conducted every 2 h

from 8:00 am to 8:00 am the next day, providing 12 cycles of measurements for each

chamber per 24 h.

2.3. Environmental variables

Various environmental variables were also measured in the lake sediment,



surface water, and atmosphere. We collected surface water and sediment samples (0–

15 cm) using a plexiglass water grab and a stainless steel sediment sampler (3 cm in

diameter) after obtaining gas samples. The water and sediment samples were

immediately stored in plastic bottles and bags, respectively. Then, all the samples

were stored in ice coolers and transported to a laboratory for analysis within a week.

In addition, we measured the wind speed at about 1.5 m above the water surface using

a portable anemometer (Testo 410-1, Testo, Germany) and the surface sediment (0–15

cm) temperature using a mercury thermometer. We used a multi-parametric probe

(556 MPS, YSI, USA) to measure the water quality factors in situ, such as electrical

conductivity and dissolved oxygen (DO) content, at each sampling site from June

2013 to June 2014. The water levels in the lake were obtained from the Xingzi

Hydrological Station, about 20 km from our sampling sites.

In the laboratory, the pH values of the water and sediment samples were

measured using a pH meter (Delta 320, Mettler–Toledo, Switzerland). Chemical

oxygen demand (COD) was measured using the spectrophotometric detection method

based on Griess reaction (Jirka and Carter, 1975; Yao et al., 2015). Chl *a*

concentration was measured via spectrophotometry (Rasilo et al., 2015; Yao et al.,

2015), which was extracted in 90% ethanol and then analyzed spectrophotometrically

at 750 and 665 nm in accordance with ISO 10 260 (1992). The SS level in the lake

water was measured by a gravimetric procedure, where the solids from the water

sample were filtered, dried, and weighed to determine the total non-filterable residue

of the sample (Fishman and Friedman, 1989). TP concentration was measured using

the molybdenum blue method after persulfate digestion (Karl and Tien, 1992; Yao et

al., 2015). In addition, the nitrate ($NO_3^-$), ammonium ($NH_4^+$), TN, and dissolved

organic carbon (DOC) contents in the water were measured using a total carbon and



nitrogen analyzer using filtered water (Shimadzu TOC-VCSH + TN module,

Shimadzu, Japan). The sediment TN and organic carbon contents after total sediment

acidification with HCl 1N were determined using a vario MAX CN element analyzer

(NA Series 2, CE Instruments, Germany).

Considering the different sampling periods, we classified the environmental

variables into three groups. The first group included sediment temperature, sediment

total nitrogen content, water level, DOC content in the water, pH in the sediment,

$NH_4^+$ and $NO_3^-$ concentrations in the water and sediment, sediment organic carbon

content, the ratio of carbon and nitrogen, and the mean daily wind speed over a

48-month period. The second group included TN, TP, COD, and Chl *a* contents in the

water, which were sampled between June 2011 and December 2014. We sampled the

third group variables from June 2013 to June 2014, including DO content,

conductivity, and pH in the water.

2.4. Data analysis

We averaged the $CH_4$ effluxes of the three sites to minimize the effect of the

spatial variation of $CH_4$ efflux on the temporal dynamics of the efflux. One-way

ANOVA followed by post-hoc Tukey's test and paired T test were used to analyze the

seasonal and inter-annual differences in the $CH_4$ effluxes. The coefficient of variation

(CV) was used to quantify the inter-annual variation of $CH_4$ efflux. We employed

stepwise multiple regressions to identify the environmental factors driving the $CH_4$

effluxes at different temporal scales. We also used regression and correlation analyses

to determine the relationships between independent variables and $CH_4$ effluxes. We

used the Vant′ Hoff equation to calculate the temperature sensitivity ($Q_{10} = e^{10b}$) of

$CH_4$ efflux (Xu and Qi, 2001; Wei et al., 2015). All statistical analyses were performed

using the SPSS 17.0 statistical software (SPSS Inc., Chicago, IL, USA), and graphs



were created using the Sigma Plot 11.0 program (Systat Software Inc., San Jose, CA,

        USA).



### 3. Results

3.1. CH$_4$ effluxes in Poyang Lake

3.1.1. Annual CH$_4$ effluxes

The mean CH$_4$ efflux was 0.54 $\pm$ 0.053 mmol m$^{-2}$ day$^{-1}$ in Poyang Lake over

the 4-year period, with annual mean effluxes of 0.47 $\pm$ 0.54, 0.56 $\pm$ 0.41, 0.52 $\pm$

0.55, and 0.60 $\pm$ 0.56 mmol m$^{-2}$ day$^{-1}$ in 2011, 2012, 2013, and 2014, respectively

(Table 1). The inter-annual variation of CH$_4$ efflux was moderately high with a CV of

9.8% over the 4 years. The mean CH$_4$ efflux in 2014 was 25.7% greater than that in

2011, justifying the necessity for long-term measurements.

3.1.2. Seasonal CH$_4$ effluxes

The seasonal variations of CH$_4$ effluxes in Poyang Lake were prominent,

demonstrating a similar pattern to that of seasonal temperature (Fig. 3). In general, the

annual maximum CH$_4$ effluxes occurred in summers and the minimum in winters. The

CH$_4$ efflux increased slowly in early spring and then rapidly in May, reaching its

maximum in July. After reaching the maximum, the CH$_4$ efflux decreased sharply in

August and September and then slowly before reaching its minimum in January (Fig.

3). Significant differences in the mean CH$_4$ effluxes existed between summers and the

other three seasons throughout the 4 years ($p < 0.05$), whereas the differences in the

CH$_4$ effluxes among the spring, autumn, and winter seasons were not statistically

significant ($p > 0.05$) (Table 1).

3.1.3. Diel CH$_4$ effluxes

The CH$_4$ effluxes in Poyang Lake also exhibited apparent variations within a day

because the daily maximum appeared late in the morning (10:00–12:00 h) and the

minimum early in the morning the next day (4:00–6:00 h). The diel pattern of the CH$_4$

efflux was asymmetric, fast increasing in the morning from 8:00 h to 12:00 h and



slowly decreasing in the afternoon and during the night, especially in the summer (Fig. 4). However, the diel pattern of the $CH_4$ efflux was inconsistent and obvious. For example, the diel pattern on January 13–14, 2013 was an exception, when the

maximum efflux occurred around 6:00 h on January 14[th] and a severe cold front with heavy fogs enveloped the Poyang Lake area in the early morning of January 14[th]. The diel pattern of $CH_4$ efflux was vague with an average difference between the daily maximum and minimum of only 0.073 mmol $m^{-2} h^{-1}$. The $CH_4$ efflux could also change abruptly throughout a day. For example, the efflux sharply dropped from

0.068 to −0.012 mmol $m^{-2} h^{-1}$ within barely 2 h, as observed on July 23, 2011, indicating that the lake switched from a $CH_4$ source to sink within a short period of time (Fig. 4a). This abrupt change was also observed in the afternoon of August 28, 2012 (Fig. 4b). Further analysis showed that the diel pattern of $CH_4$ effluxes followed the diel pattern of wind speed (Figs. 5a–5d).

3.2. Relationships between $CH_4$ efflux and environmental variables

In the current study, environmental factors differed in importance depending on the timescale in the stepwise multiple regressions analyses. The results of stepwise multiple regressions on a seasonal scale showed that the sediment temperature, sediment TN content, DO, and TP content in the water were significant predictors of

$CH_4$ effluxes (Table 2). In specific, sediment temperature and sediment TN content explained 65% of the variation in $CH_4$ effluxes for 4 years when we used the first group of factors. The sediment temperature and TN content explained 73% of the $CH_4$ efflux variations when the second group of variables was added to the first group. The sediment temperature, sediment TN content, DO, and TP contents in the water

explained 89% of the $CH_4$ efflux variation when the three groups of variables were used together. Wind speed was the only significant variable for the $CH_4$ efflux





variations on a diel scale. Wind speed explained 58%, 56%, 84% and 86% of the $CH_4$

efflux variations in 24–25 July 2011, 5–6 September 2012, 13–14 January 2013 and

14–15 January 2015, respectively (Fig. 5a-5d).






## 4. Discussion

### 4.1. $CH_4$ effluxes in Poyang Lake

The mean $CH_4$ emission in Poyang Lake was moderately higher than those in other

large lakes of more than 1 $km^2$ in the world. The mean $CH_4$ emission (0.54 mmol $m^{-2}$

$day^{-1}$) was within the reported range of approximately 0.022–5.85 mmol $m^{-2}$ $day^{-1}$ in

boreal and temperate lakes over 1 $km^2$ but was obviously lower than diffusive effluxes

in subtropical lakes and total effluxes (including diffusion and ebullition) in tropical

lakes (Table 3). In addition, the mean $CH_4$ emission in Poyang Lake was comparable

with the diffusive effluxes in tropical lakes (Table 3). For example, previous studies

reported that the diffusive $CH_4$ efflux was 0.65 mmol $CH_4$ $m^{-2}$ $day^{-1}$ in the TR Lake

and 0.50 mmol $m^{-2}$ $day^{-1}$ in the BB Lake in the Pantanal region (Bastviken et al.,

2010). However, the mean $CH_4$ efflux in Poyang Lake was only higher than those in

other lakes over 100 $km^2$ (except the Võrtsjärv Lake). The low $CH_4$ efflux in the

current study was unlikely caused by our floating chamber system because the $CH_4$

efflux would have increased if the insertion of chambers considerably disturbed the

water profiles. The lower $CH_4$ emissions in our study may be attributed to the low

concentration of carbon substrates in the water and sediments in Poyang Lake. The

DOC concentration in Poyang Lake was merely 3.3 mg $L^{-1}$, which was much lower

than that of the 5.8 mg $L^{-1}$ in Biandantang Lake and 7.4 mg $L^{-1}$ in Donghu Lake,

which are two subtropical lakes in China (Xing et al., 2005, 2006). Poyang Lake also

has a lower organic carbon content in its sediments than most other lakes. The average

organic carbon content in the sediments in Poyang Lake was 0.89%, which was much

lower than that of 30.76% averaged over five temperate lakes (Schrier–Uijl et al.,

2011) and slightly higher than that of nearly 0.75% in tropical lakes in the Pantanal

region (Bastviken et al., 2010). Therefore, the $CH_4$ emissions in large lakes cannot be



ignored when estimating the global $CH_4$ budget because of their area.

CH$_4$ effluxes at the air-water interface showed high fluctuations in the four cycles, but showed no significant diurnal differences. The diurnal $CH_4$ efflux ranged from -0.019 to 0.13 mmol m$^{-2}$ h$^{-1}$, which was within the reported range of other lakes (-0.057 to 5.37 mmol m$^{-2}$ h$^{-1}$) over a diurnal cycle (Xing et al., 2004; Duan et al., 2005; Chen et al., 2007; Podgrajsek et al., 2014a, 2014b). The wide range of diurnal $CH_4$ efflux in previous results may be due to differences in sample size in different studies. For example, CH$_4$ efflux was measured at 2h intervals with 12 data points over a diurnal cycle in this study, but $CH_4$ efflux was measured at 3-6 h intervals with only 4-8 data points in previous studies (Käki et al., 2001; Xing et al., 2004; Duan et al., 2005). Another possible reason for these discrepancies was that vegetation might have played an important role in $CH_4$ efflux in other studies (Käki et al., 2001; Duan et al., 2005), while there was no vegetation in the water where we sampled in Poyang Lake. In addition, our study showed that there were no significant differences in $CH_4$ effluxes between the nighttime and the daytime, which was inconsistent with other studies (Keller and Stallard, 1994; Bastviken et al., 2010). This inconsistency may be due to incomplete measurement of the diurnal cycle in other studies. For example, CH$_4$ efflux was only measured three times at sunrise, daytime and sunset to represent a diel cycle in Bastviken et al. (2010). In Keller and Stallard (1994) study, the daytime and nighttime $CH_4$ efflux measurements were not conducted on the same day. In particular, two studies reported a new finding that hydrodynamic transport contributed more to nighttime $CH_4$ effluxes than daytime $CH_4$ effluxes (Poindexter et al. 2015; Anthony and Macintyre 2016). However, we cannot estimate $CH_4$ effluxes by hydrodynamic transport because we did not measure $CH_4$ concentration in the water in this study. Further studies are needed to address this issue in the lake.



4.2. CH$_4$ effluxes in summer

The CH$_4$ effluxes in Poyang Lake were substantially greater in summer than in the other seasons, accounting for more than 63% of the annual total emissions. This finding suggests that summer is the critical season in managing the CH$_4$ emissions from Poyang Lake. The high effluxes in summer may be attributed to the higher temperature, higher substrate availability, and greater temperature sensitivity during this season than the other seasons.

Poyang Lake features a typical monsoon climate with hot summers. During the study period, the mean (June–August) air temperature in summer was 28.5 ℃, whereas that in winter was only 5.9 ℃. The CH$_4$ effluxes were highly correlated with the sediment temperature through an exponential function. Our results confirmed the findings of previous studies that lake CH$_4$ effluxes are driven by temperature (Bastviken et al., 2008; Marinho et al., 2009; Palma–Silva et al., 2013; Rõõm et al., 2014). This is supported by the fact that a warm temperature provides a high optimal temperature for methanogen growth, which increases methane production (Nozhevnikova et al., 2007; Rooney–Varga et al., 2007; Duc et al., 2010). Moreover, recent studies have reported that high temperatures could increase the proportion of hydrogenotrophic methanogenesis, which is an important pathway for CH$_4$ production (Borrel et al., 2011; Marotta et al., 2014). The high summer CH$_4$ effluxes might also be because of the ample substrate supply in this season. In the present study, CH$_4$ efflux positively correlated with the Chl $a$ content (r = 0.46, data not shown) that was not correlated with other environmental factors and acted as an indicator of primary production. Earlier studies discovered a high amount of labile organic matter, including allochthonous inputs of terrestrial organic matter, during the summer flooding and autochthonous production within-lake by phytoplankton and benthic




algae in summer (Crump et al., 2003; Xing et al., 2005, 2006; Bade et al., 2007). The

decomposition rate of new organic matter was much faster than that of old organic

matter (Davidson and Janssens, 2006; Gudasz et al., 2010). Previous studies showed

that fresh organic carbon from dead algae stimulates $CH_4$ emissions in lakes

(Huttunen et al., 2002; Xing et al., 2005) because the degradation of dead alga and

algal exudates, such as methylated compounds, are the precursors for $CH_4$ production

(Ferrón et al., 2012; Xiao et al., 2015; Liang et al., 2016). However, we did not find

any correlation between the $CH_4$ efflux and DOC content in the water ($p > 0.05$). The

algal bloom in summer probably masked the DOC effect on stimulating $CH_4$

production. Earlier studies demonstrated that 70%–80% of DOC molecules in lakes

are recalcitrant carbon, which are composed of humic substances in the lake from the

partial degradation of terrestrial lignin in vegetation (Tranvik and Kokalj, 1998;

Wetzel, 2001).

The high summer $CH_4$ effluxes were also driven by the greater temperature

sensitivity during summer. The apparent $Q_{10}$ value in Poyang Lake was 2.04 in

summer, which was much greater than the value of 1.67 in the other seasons (Fig. 6).

This finding is inconsistent with previous studies in terrestrial and freshwater

ecosystems (Davidson and Janssens, 2006; Gudasz et al., 2010; Yvon–Durocher et al.,

2014), where the $Q_{10}$ values decreased apparently with the increase in temperature

(Xu and Qi 2001a; Chen et al., 2010; Corkrey et al., 2012; Schipper et al., 2014).

However, our result was supported by a recent finding that the temperature

sensitivities ($Q_{10}$) of $CH_4$ effluxes from lake sediments are greater in the tropics than

in boreal regions (Marotta et al., 2014). We speculate that the temperature effect on

$Q_{10}$ was confounded by other factors, such as water level and substrate availability.

The addition of a large amount of fresh carbon from summer floods could



dramatically boost $CH_4$ production and thus the apparent $Q_{10}$ values during summer.

4.3. Timescale dependence of wind, substrate availability, and temperature effects on $CH_4$ effluxes

In this study, the effects of wind, substrate availability, and sediment temperature on $CH_4$ effluxes were highly timescale dependent. The $CH_4$ effluxes measured at bihourly intervals positively correlated with wind speed in both simple and multiple regressions (Figs. 5a–d, Table 2) but showed no correlation ($p > 0.05$) when the diurnal or seasonal average $CH_4$ efflux and wind speed were applied (Figs. 5e–f). The

effect of wind on $CH_4$ effluxes was mainly through its effects on the transport, air pressure and storage of $CH_4$ from the bottom to the surface water (Abril et al., 2005; Hahm et al., 2006; Gu érin et al., 2007). Gas diffusion in water is sensitive to pressure changes at the water–air interface (Paganelli et al., 1975; Massmann and Farrier, 1992; Striegl et al., 2001; Nachshon et al., 2012). High wind speed mechanically induces

turbulences through friction in the water and brings $CH_4$-rich water from the bottom to the surface in lakes (Wanninkhof, 1992; Palma–Silva et al., 2013; Xiao et al., 2013). The $CH_4$ efflux rapidly decreases or even becomes negative (indicating $CH_4$ absorption) to compensate for the deficits in the water profile caused by earlier winds when the wind declines or comes to a halt. Our results also confirmed that the $CH_4$

efflux sharply declined to a negative value after strong wind events (Fig. 4). This wind effect only worked at short timescales, such as bihourly, when temperature only slightly changed and other biological processes, such as microbial community variation, were relatively stable. At a longer temporal scale, such as seasonal scale as observed in the current study, the wind effect disappeared because the

wind-stimulated $CH_4$ effluxes and the post-wind (or between-gusts) negative effluxes (absorptions) were compensated. Our results suggest that wind exerts minor effects on





CH$_4$ effluxes at large temporal scales when temperature, water level, and substrate availability dominate. Our results also suggest that caution must be taken when one applies the empirical wind speed-driven models developed based on short-term
measurements to estimate CH$_4$ effluxes over long periods, such as months or years.

Meanwhile, the CH$_4$ effluxes measured at monthly intervals positively correlated with sediment temperature (Fig. 6, Table 2), but the correlation disappeared when applied at bihourly intervals (p > 0.05). The lack of correlation between the CH$_4$ efflux and sediment temperature as measured on a bihourly scale within a day can be
explained by the small variation of sediment temperature within a day, ranging from 0.95 ℃ to 1.85 ℃. Other factors, such as wind and atmospheric pressure, might shadow the weak temperature effect within a day. Instead, we found a high correlation between the bihourly measured CH$_4$ effluxes and sediment temperature during the diel measurement period in January 14 to 15, 2015 (r = 0.88, p < 0.0001). Further analyses
showed that this temperature effect might be apparent and mainly caused by wind speed because the bihourly measured CH$_4$ effluxes and wind speed were highly correlated only in January 14 to 15, 2015 and not in the other days (r = 0.90, p < 0.0001). However, sediment temperature became the dominant factor on a seasonal scale when the temperature ranged from about 4.4 ℃ in winter to 30.8 ℃ in summer
(Fig. 3). The sediment temperature and CH$_4$ effluxes averaged over the diurnal period significantly correlated in the 4-year study period (Fig. 6, Table 2). Our results suggest that the short-term CH$_4$ efflux in Poyang Lake was regulated by wind speed, but the long-term CH$_4$ efflux was ultimately controlled by sediment temperature and other biological (e.g., microbial activities) and biochemical (e.g., sediment carbon and
nitrogen contents) processes. Therefore, understanding and modeling the dynamics of CH$_4$ effluxes on lake surfaces require the long-term measurements of effluxes and





related biotic and abiotic factors in lake water and sediments. Finally, substrate availability, such as sediment TN content, TP, and Chl *a* contents in the water, also influenced $CH_4$ effluxes on a seasonal scale in the current study (Table 2). However,

the effects disappeared when applied at bihourly intervals because the substrate did not change significantly within a day.

In addition to the above-mentioned factors, the DO concentration in the water influenced the $CH_4$ effluxes in the multivariate regression analysis. In specific, the $CH_4$ efflux closely correlated with the DO concentration in the water (r = –0.65). This

close correlation can be explained by the aerobic $CH_4$ oxidation in the water. Our result was supported by the previous finding that a high DO concentration in the water results in low $CH_4$ emission (Rõõm et al., 2014; McNicol and Silver, 2015; Yang et al., 2015).



**5. Conclusion**

The average $CH_4$ efflux in Poyang Lake during the 4-year study period was 0.54 $\pm$ 0.053 mmol m$^{-2}$ day$^{-1}$, which was moderately higher than that of the other lakes in the world. The $CH_4$ efflux in Poyang Lake also featured high seasonal variations with the maximum efflux in July and the minimum in January. About 63% of the annual

emissions occurred in summer, from June to August. On a seasonal scale, multivariate regression analyses revealed that sediment temperature sediment TN content, TP, and DO contents in the water mainly regulated the $CH_4$ effluxes. Simple and multivariate regression analyses showed that wind speed influenced the diel $CH_4$ efflux variations. The effects of sediment temperature, substrate availability, and wind speed on $CH_4$

effluxes were temporal scale dependent. The $CH_4$ effluxes increased with the sediment temperature, sediment TN content, Chl *a*, and TP contents in the water on a seasonal scale but were not correlated with sediment temperature on a bihourly scale. In contrast to the temperature and substrate, the $CH_4$ efflux positively and significantly correlated with wind speed within a day on a bihourly scale but was not

correlated with wind speed at larger temporal scales, such as daily and seasonal scales. The timescale dependence of environmental controls on $CH_4$ effluxes has important implications in modeling $CH_4$ emissions.





**Acknowledgements**

This research was supported by the National Basic Research Program of China (973 Program, 2012CB417103), Assessment and Valuation of Ecosystem Services in Qinghai Provincem, China (No. 2013-N-556). We gratefully acknowledge the Poyang Lake Laboratory Wetland Ecosystem Research, CAS for permission to access the study site and assistance with field work.






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



Table 1 Seasonal and annual means of CH$_4$ effluxes with the chamber measurements in Poyang Lake

| CH$_4$ efflux (mmol m$^{-2}$ day$^{-1}$) | 2011 | 2012 | 2013 | 2014 |
|---|---|---|---|---|
| Spring (Mar–May) | 0.22 ±0.035b | 0.36 ±0.092 bc | 0.23 ±0.16b | 0.37 ±0.084 b |
| Summer (Jun–Aug) | 1.34 ±0.31a | 1.21 ±0.16a | 1.36 ±0.44 a | 1.44 ±0.46a |
| Autumn (Sep–Nov) | 0.23 ±0.12b | 0.43 ±0.14b | 0.33 ±0.12b | 0.34 ±0.16 b |
| Winter (Dec–Feb) | 0.11 ±0.014b | 0.23 ±0.036 b | 0.14 ±0.047 b | 0.23 ±0.10 b |
| Mean | 0.47 ±0.54a | 0.56 ±0.41ac | 0.52 ±0.55a | 0.60 ±0.56bc |

**Note**: Means with different letters are significantly different as determined by multiple comparisons on a seasonal scale (one-way ANOVA, post hoc Tukey test, $p < 0.05$) and a pair T
test ($p < 0.05$) on an annual scale.


**Table 2** Multivariate regressions between seasonal CH$_4$ efflux and environment factors

| No. | Number of variables | Regression Equation | n | $R^2$ | p |
|---|---|---|---|---|---|
| Group 1 | 12 | EffluxCH$_4$ = −10.48 + 110.57 ST + 65.06SN | 48 | 0.65 | 0.004 |
| Group1 + Group 2 | 16 | EffluxCH$_4$ = −12.66 + 0.57ST + 90.81SN | 43 | 0.73 | 0 |
| Group 1 + Group 2 + Group 3 | 19 | EffluxCH$_4$ = −3.89 + 0.56ST + 102.88SN − 35.56TP − 0.74DO | 19 | 0.89 | 0 |

Note: Nd means that no variable input to the stepwise regression exists. Variables in group 1 included sediment temperature, sediment

total nitrogen content, water level, DOC content in the water, pH in the sediment, NH$_4^+$ and NO$_3^-$ concentrations in the water and in the

sediment, sediment organic carbon content, the ratio of carbon and nitrogen, and the mean daily wind speed. Variables in group 2 included

TN, TP, COD, and Chl *a* contents in the water. Variables in group 3 included DO content, conductivity, and pH in the water.





Table 3 Mean $CH_4$ effluxes in Poyang Lake in comparison with other large lakes

| Lake | Lake size ($km^2$) | Region | Climate | $CH_4$ efflux (mmol $m^{-2}$day$^{-1}$) | References |
|---|---|---|---|---|---|
| 11 lakes | 1 | Laurentians, Canada | Boreal | 4.08 | Rasilo et al., 2015 |
| 19 lakes | 47 | Chicoutimi, Canada | Boreal | 1.08 | Rasilo et al., 2015 |
| 21 lakes | 41 | Abitibi, Canada | Boreal | 1.67 | Rasilo et al., 2015 |
| 14 lakes | 171 | Chibougamau, Canada | Boreal | 0.17 | Rasilo et al., 2015 |
| 14 lakes | 7 | James Bay, Canada | Boreal | 1.08 | Rasilo et al., 2015 |
| 45 lakes | 5 | C^ote-Nord, Canada | Boreal | 1.17 | Rasilo et al., 2015 |
| 14 lakes | 2 | Eastmain, Canada | Boreal | 0.58 | Rasilo et al., 2015 |
| 48 lakes | 242 | Scheffervill, Canada | Boreal | 0.42 | Rasilo et al., 2015 |
| Lake Mendota | 39.4 | North America | Boreal | 0.50 | Fallon et al.,1980 |
| Erie | 25700 | North America | Boreal | 0.04 | Howard et al.,1971 |
| Dillon | 13 | North America | Boreal | 0.61 | Smith and Lewis,1992 |
| Fiolen | 1.5 | Sweden | Boreal | 0.02 | Bastviken et al., 2004 |
| Kevätön | 4 | Finland | Boreal | 0.22 | Huttunen et al., 2003 |
| Biwa | 674 | Japan | Temperate | 0.27 | Miyajima et al.,1997 |
| Constance | 540 | Europe | Boreal | 0.04 | Schultz et al., 2001 |
| Kasumigaura | 168 | Japan | Temperate | 0.26 | Utsuumi et al.,1998a |
| Nojiri | 4.4 | Japan | Temperate | 0.06 | Utsuumi et al.,1998b |
| 5 lakes | range 1–11, 3436[A] | Netherlands | Temperate | 5.85 | Schrier–Uijl et al., 2011 |
| Donghu | 27.9 | China | Subtropical | 1.46 | Xing et al., 2005 |
| TR lake | 71.4 | Pantanal, South America | Tropical | 0.65[B]/5.74[C] | Bastviken et al., 2010 |
| BB lake | 36.3 | Pantanal, South America | Tropical | 0.50[B]/5.63[C] | Bastviken et al.,2010 |
| Biandantang | 3.3 | China | Subtropical | 1.32 | Xing et al., 2006 |
| Võrtsjärv | 270 | Estonia | Boreal | 1.28[B]/2.09[C] | Rõõm et al., 2014 |





| 43 lakes | range1–10, 782073.8[A] | worldwide | Mainly boreal | 0.12 | Holgerson and Raymond, 2016 |
|---|---|---|---|---|---|
| 18 lakes | range10-100, 597789.3[A] | worldwide | Mainly boreal | 0.10 | Holgerson and Raymond, 2016 |
| 6 lake | >100, 2024015.8[A] | worldwide | Mainly boreal | 0.06 | Holgerson and Raymond, 2016 |
| Poyang Lake | 3283 | China | Subtropical | 0.54 | Present study |

Note: A means total areas in the given lake size. B means diffusive effluxes and C means total effluxes, including diffusion and ebullition.




**Figure Captions**

Figure 1. Location of sampling sites in Poyang Lake.

Figure 2. Examples of calculating the slope of total effluxes, including diffusive and ebullitive

effluxes. All the concentrations are presented in original (volumetric parts per million-units).
White circles represent the $CH_4$ concentrations at different sampling times. Grey circles
represent the adjusted concentration. Black trendlines represent the data used for the total
efflux calculation. The different letters in the figure panels mean different occurrence times
for ebulltion: no ebullition (a), occurrence of ebullition at 20 min (b), 40 min (c), and 60 min

(d), respectively.

Figure 3. Seasonal variations of $CH_4$ effluxes and sediment temperatures in Poyang Lake.
White circles represent the variation of $CH_4$ effluxes, and black circles describe the variation
of sediment temperature in the 4-year period.

Figure 4. Diel variations of $CH_4$ effluxes in Poyang Lake.

Different panels present the diel variations of the $CH_4$ effluxes in 24–25 July 2011 (a), 5–6
September 2012 (b), 13–14 January 2013 (c), and 14–15 January 2015 (d). White circles
describe the diel variations of the $CH_4$ effluxes. Horizontal short dashed lines mean the
average value of the diel $CH_4$ effluxes.

Figure 5. Relationships between $CH_4$ effluxes and wind speed in Poyang Lake.

White circles represent the observed values of $CH_4$ effluxes and wind speed. Different panels
mean the variations of $CH_4$ effluxes at a bihourly interval within a day, including in 24–25
July 2011 (a), 5–6 September 2012 (b), 13–14 January 2013 (c), and 14–15 January 2015 (d),
on a diurnal scale (e), and on a seasonal scale (f). Panels e and f include all the measurements
during the observation period. We excluded the white-crossed circle in figure c in the

regression analysis because of a severe cold front.

Figure 6. Relationship between sediment temperature and $CH_4$ effluxes in Poyang Lake.
White circles represent the observed values of the diurnal mean $CH_4$ effluxes and sediment
temperature in summer, and black circles represent the observed values of the diurnal mean



CH$_4$ effluxes and sediment temperature in the other seasons in the 4-year period. Black lines

represent the fitting curves of the relationship between CH$_4$ effluxes and sediment

temperature.





Fig. 1


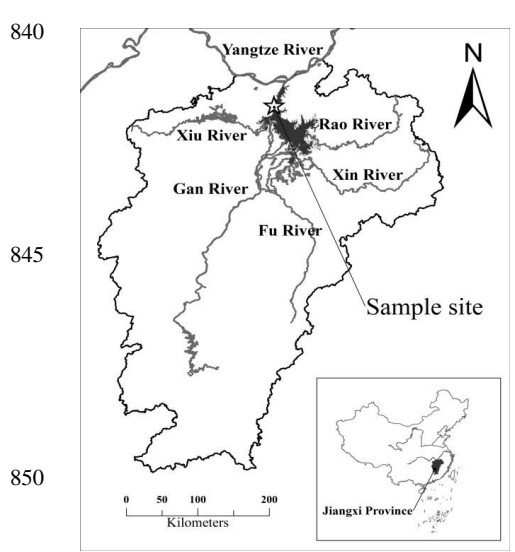

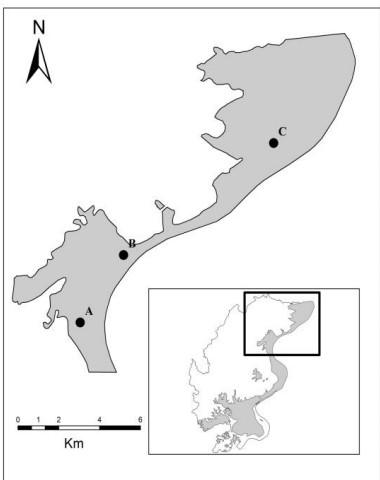










Fig. 2

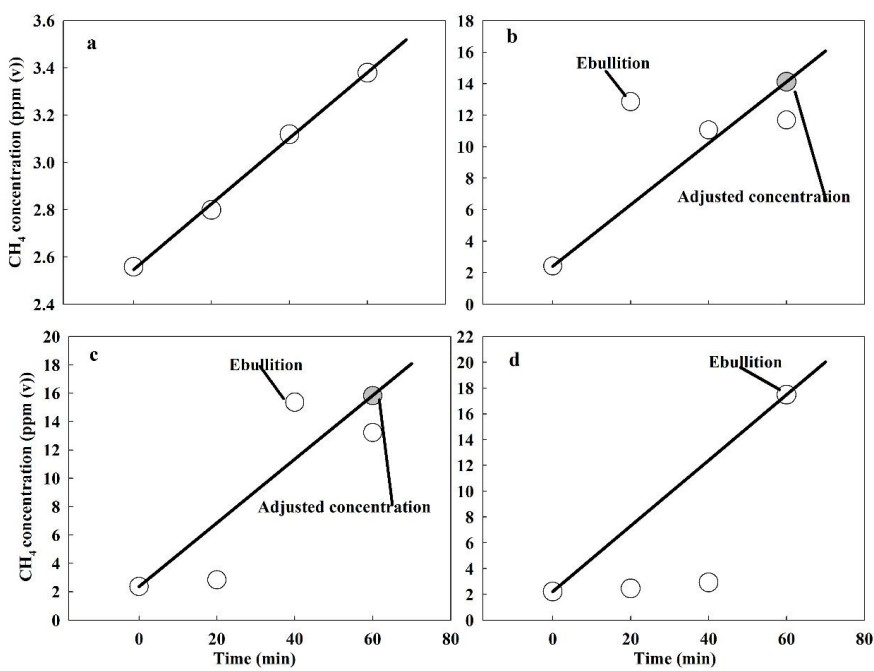




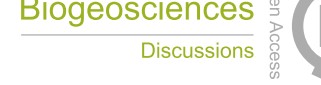

Fig. 3


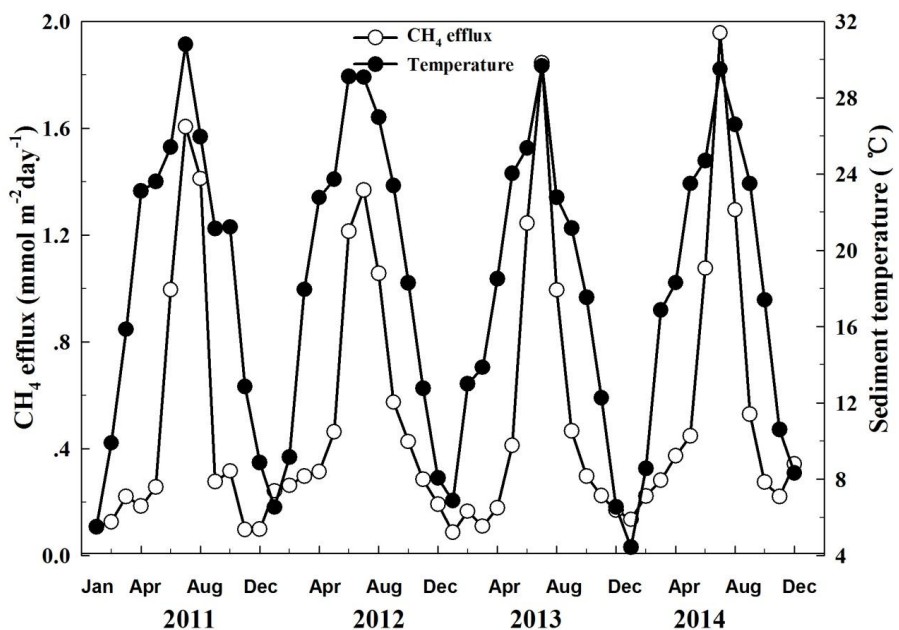








Fig. 4

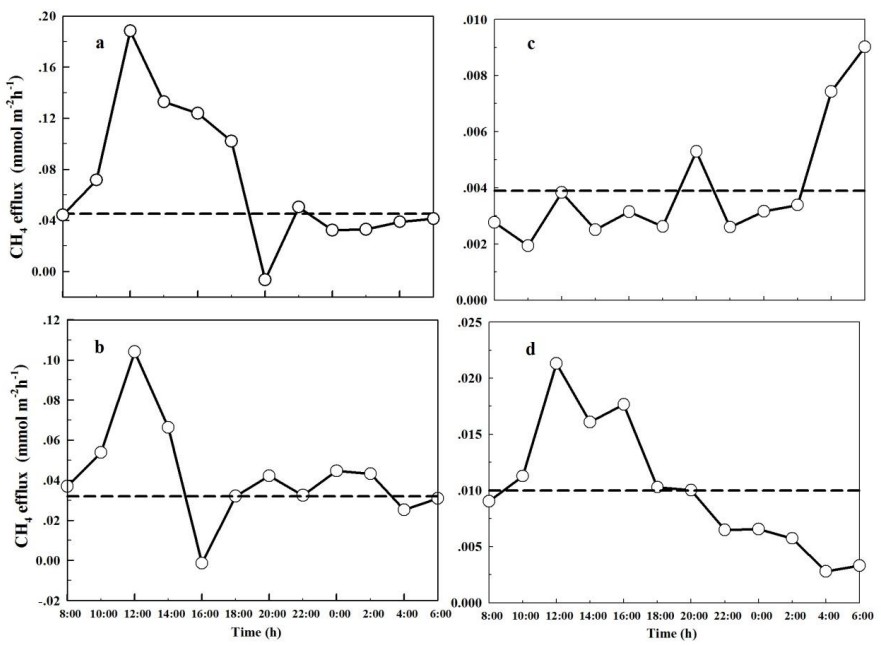




Fig. 5

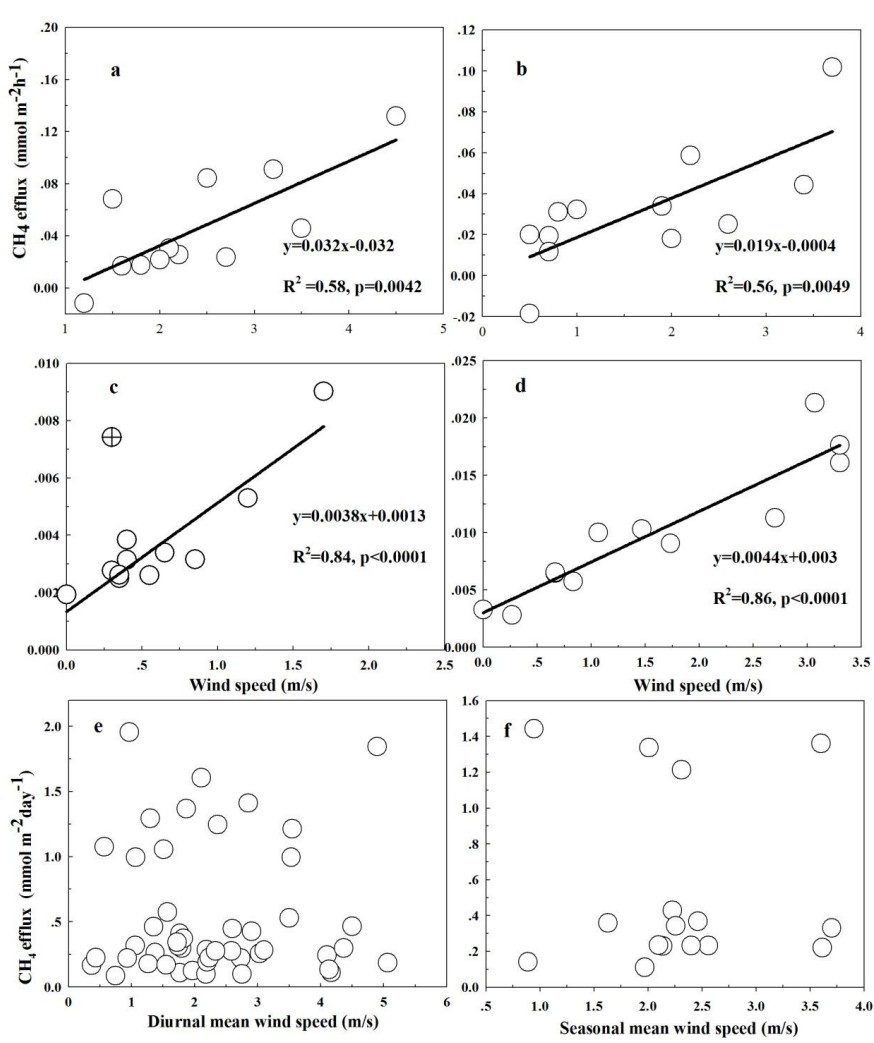







Fig.6

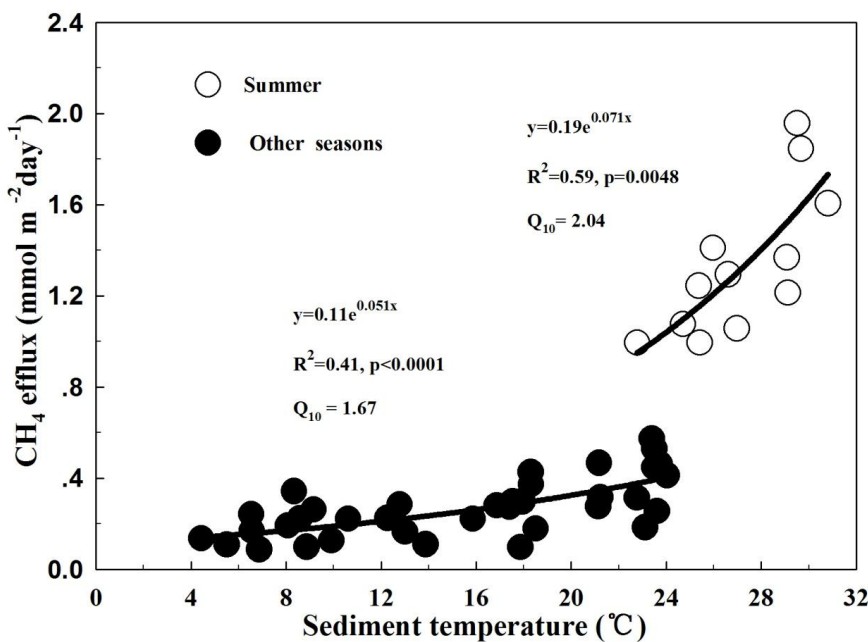
