# Peer review of "Timescale dependence of environmental controls on methane efflux in Poyang Lake, China"

_Biogeosciences, 2016_

## Referee Comment (RC1) · Anonymous Referee #1 · 22 Sep 2016

General comments

The manuscript reports on the temporal patterns of methane ($CH_4$) efflux in the largest lake in China and the various factors that influence these fluxes over different timescales. $CH_4$ efflux was slightly greater than other lakes with an area greater than 1km, but was comparable to that found in tropical lakes. The variables best explaining variation on $CH_4$ efflux was timescale dependent but, overall, temperature was important over seasonal scales and wind speed on a bihourly scale.

The paper is well written despite a few grammatical errors. As the authors point out, there is a lack of data explaining $CH_4$ effluxes in this region and also in larger, non-alpine lakes more generally. As such, the results from this paper will add to the limited understanding of $CH_4$ dynamics in these lake types. However, I cannot recommend

that this paper be published in its current form. I have major issues with a) the premise of the paper, b) some overreaching statements that are made, and c) the statistical approaches used - all have major implications for the generalisation of the results. It is difficult for me to assess the results and technical aspects of this study until statistical changes are made.

One major concern is that the study was undertaken in a very small area (three sites with 20km of each other) even though the lake is the largest in China by area (3283 sq km). Further, the study sites are situated in a section of the lake that appears to be relatively confined. There is nothing wrong with the site selection. However, the authors cannot make statements about the whole lake because they don't know if the spatial and temporal patterns of CH4 vary the same way across the lake. They need to qualify in all statements that the research was undertaken in one small section of the lake. It is not a study of CH4 effluxes from Poyang Lake, but it is a study of CH4 effluxes from one section of Poyang Lake.

Another major concern is a statistical one. The authors use average values from three different locations in Poyang Lake for all analyses. The justification for this was to 'minimize the effect of the spatial variation of CH4 efflux on the temporal dynamics of the efflux'. However I suspect the main motivation for doing this was because the environmental variables were only collected at one location (it is not clear where the environmental variables were collected). Was this the case? Given that CH4 was only measured in three locations of the lake, surely the degree of variation between them is very important to a) understand and/or b) account for in statistical analyses. The authors should re-analyse their results in one of the following ways: • Treat each study site as a random effect in mixed effects models so that variation among the three sites in taken into account when investigating the annual, seasonal, and diurnal variation, as well as the relationships with measured explanatory variables. Including site as a random effect would enable the researchers to make more general statements about CH4 fluxes from Poyang Lake – this is just common practice these days and

should be incorporated into the study design / statistical analyses. A random effect for site effectively means that these study sites are a random sample of all potential sites in the lake – this is where the generalisability comes in. Please see Section 8.1.1 (Types of predictor variables (factors)) in Quinn & Keough (2002; Experimental Design & Data Analysis for Biologists) or another similar book for information about mixed effects models and random and fixed factors. • Split the analyses into two parts. The first analysis will not average the three study sites prior to the analysis and investigate the spatial and temporal patterns in CH4 among them. The second analysis could average the study sites (still preferably treat study site as a random factor) and relate this to the measured explanatory variables.

One more major concern is the notion that this is a long-term study. 4 years is not long term. Remove all reference to this study being long term, including the second sentence of the Abstract which introduces the idea that this research is filling the knowledge gap around the lack of long term research on CH4 fluxes. Instead, the authors should frame this 'knowledge gap' around the lack of multi-seasonal investigations into CH4 effluxes – this is exactly what this paper addresses.

Specific comments

Line 18. It is stated continuous measurements of CH4 efflux was measured, but measurements where not continuous. Monthly measurements were made. Change all reference to continuous measurements in the manuscript to monthly measurements. Line 121-124. Are these parameters an average of the entire lake or for a specific location? Please specify. Section 2.3. Environmental variables. Where were the environmental variables collected from? Where samples collected at each of the three study sites and then averaged or from just one site? This information is very important. Line 331-332. This concluding sentence only relates to the first sentence of this paragraph and does not relate or link to the remaining text in the paragraph. This sentence should only be left if a re-working of the paragraph better supports this argument. Line 337-341. An argument is made that this study has lower diurnal variation in CH4 efflux than other

studies and this may be due to differences in sample size in other studies. I would think that more frequent sampling would in-fact lead to more variation. The authors need to report on how much diurnal variability in CH4 efflux there was among the study sites.

Technical comments

Line 163-182. The description of how CH4 efflux due to ebullition is very confusing and long. From Line 125, where the ebullition and diffusive fluxes are introduced, I would suggest briefly describing how, or how not, the chambers can be used to differentiate these two fluxes. Line 312. Remove 'obviously'.

---

## Referee Comment (RC2) · Anonymous Referee #2 · 26 Sep 2016

This manuscript presents 4 years of CH4 flux patterns in the largest lake in China and environmental factors that influence CH4 flux rates. It falls well within the scope of Biogeosciences, but several aspects need to be improved for publication.

Some suggestions: 1) How do you define "long-term"? To me, 4-year observations can be short-term. Also, all the statements related to seasonal or interannual variability need to be justified because CH4 flux rates measured on one day may not represent flux rates of one month. Furthermore, daily CH4 flux rates could have been overestimated, considering that CH4 flux rates are measured during the day each month, when CH4 flux rates were higher than those at night according to diel cycle measurements.

2) All the assumptions are met for regression models? Did you consider any interactions among variables? In addition, did you also carry out the analysis before averaging

the flux rates, with replicates as random effects? If so, how did the results differ from those after averaging?

3) In the discussion section some results were described, which did not appear in the result section. Results and discussion need to be better separated. In addition, the interpretation of the results needs to be better supported in the discussion section, focusing clearly on the core messages, i.e., what the results mean and what we can learn from this study.

Line#47-51, there are too few references to represent the minimum and maximum flux rates in lakes, especially given that those references are from lakes in China and Norway only. Also, if such values can be presented with more studies, how would seasonal variations look like in comparison to diurnal ones?

Line#75-78, can you add references for each variable? Line#64-72 well covered the references for each variable, but this section lacks it.

Line#78-82, it sounds like investigating in large lakes is not important. Please rephrase or add some more sentences to justify the importance of this research.

Line#86-87, I suggest adding references that describe the previous studies, e.g. Liu et al. (2013).

Line#109, what are the species names of Carex?

Line#128-146, this section can be written more concisely.

Line#166, can decreases in CH4 concentrations right after ebullition events be solely explained by diffusion back to lake water? If CH4 molecules were diffused back to the lake water, partial pressure of CH4 inside the chamber should be very high, inhibiting further emission from lake water to chamber. Can they be partially from irregular air mixing inside the chamber, which results in errors in CH4 concentrations? Then, the current method for calculating flux rates needs to be reconsidered.

Line#167-182, this section is confusing. It can be written clearly and concisely.

Line#200, were water and sediment samples collected at three sampling points for flux measurements?

The paragraph from line#229 can be given in a Table.

Line#241, T test → t-test

Line#242, flux rates are measured three times per season and they may not well represent flux rates of one season of the year. Then, can deviation of these three values be used to quantify interannual variability?

Line#247, please write what b represents in the equation.

Line#278, what do you mean by "inconsistent and obvious"?

Line#309-331, this part can be written more concisely.

Line#331-332, sentences of this paragraph do not support this conclusion.

Line#335, here again, can the absolute values be compared with a few references, which are probably based on different observation periods?

Line#338-342, a larger number of data points can produce wider range of values.

Line#345-356, possible explanations can be added, such as potential drivers that can affect diel CH4 flux patterns and their variations (if measured).

4.2 CH4 effluxes in summer, this section contains a lot of new results, which were not presented in the result section. Also, some sentences describe very detailed information from other studies, which hinders the main focus of the paragraph.

4.3 Timescale dependence of wind, substrate availability, and temperature effects on CH4 effluxes, here again, a lot of new results are reported, such as line#410-414, line#436-451, line#457-461 (repetition from result section), and line#462-468.

Line#473-475, considering uncertainties related to infrequent measurements (CH4 ef-flux rates measured on one day may not represent the mean rates of that month), this kind of statement needs to be corrected.

Table 3, can you add the observation period of each study for better comparison? Also, sorting the rows by lake size and climate would make this Table easier to read.

Figure 3 and 4, can you add error bars from spatial variability?

---

## Referee Comment (RC3) · Anonymous Referee #3 · 20 Oct 2016

Review: Timescale dependence of environmental controls on methane efflux in Poyang Lake, China.

Authors: Liu et al.

The present manuscript centers on relating methane efflux in Poyang Lake, China to the timescale dependence of environmental parameters. The study uses statistic methods to correlate methane efflux measured by floating chambers with other environmental factors under shorter and longer timescales. The results of measurements throughout the 4 years showed that temperature is an important factor controlling seasonal changes of methane efflux. Wind speed has negligible effects on methane efflux on daily and seasonal scales. The objectives fall well to the goal of Biogeosciences. However, the results and discussions did not show strong relevance to the Biogeosciences

so that I can't recommend publication of this manuscript in Biogeosciences.

Specific Comments:

1. Most of the results and discussions were built on the environmental variables and methane flux data. However, there are no data of biogeochemical related environmental variables shown in the figures and tables except Table 2. I would suggest to present the raw data of measured environmental variables in the supplementary material.

2. Substrate availability (Line, 432), biological (e.g., microbial activities) and biochemical (e.g., sediment carbon and nitrogen contents processes) (Lines 454-455) are very important factors to link methane efflux to the biogeochemical cycles and understand methane source and sink. Unfortunately, no comprehensive data or evidence to support the role of substrates and microbial activities on methane efflux in this manuscript which could be an important contribution to this journal.

3. It might be a risk to use the data from three sampling sites measured from one day (1 hr? Line 148) to represent methane efflux in that month. For example, it appears a contradiction between high methane efflux measured in July 2011 in Figure 3 and low methane efflux measured in July 2011 in Figure 4a.

4. How long and what time did the authors deploy the floating chambers in the three sampling sites within a day for the study at the large temporal scales (Fig. 3)? I feel 4-year measurements are not a very large temporal scale especially there are no continues measurements/monitering such as deploying floating chamber within a short interval (every week or every two to three days). Since high methane efflux was shown in the early mornings in Fig. 4a, b and d, were the floating chambers deployed at the same time at three different sites for the data shown in Fig. 3?

5. The area and water table of Poyang Lake fluctuate dramatically between the wet and dry seasons. The authors only have short but not clear descriptions of the effect of water level on methane efflux, e.g., in Lines 404-405 and Line 432. Methane efflux

might be high in dry seasons instead of summer, since methane efflux is expected to be high under lower water level due to decreasing of the hydrostatic pressure (e.g. Chanton et al. 1989). Are there any difference in water level between three sampling sites in different seasons (The mean water depth at three sites should not be always 3m through the whole year; Line 186)? The authors might consider a simple calculation of methane solubility changes due to water level fluctuations to strength the role of water level on methane efflux, e.g., Line 432.

6. As the authors stated in the introduction that methane is driven by three major mechanisms such as molecular diffusion, bubble ebullition and plant-mediated transportation, bubble ebullition is not the only pathway for methane to transport from water to the air. However, data for dissolved methane concentrations in lake water and sediments are lack in this study. No bubble ebullition doesn't mean no methane efflux. I would suggest to include diffusive methane flux to the air for comparison in the future by analyzing surface water methane concentrations and using the equation from the gas-transfer model e.g., Wanninkhof (1992).

7. Since many environmental factors and methane fluxes collected in October 2010 in Poyang Lake have been shown in Liu et al., (2013) for spatial studies, the authors may include Liu et al. (2013) in the introduction and discussions to emphasize why the three sampling sites were chosen in this timescale study and the relations between different environmental factors and methane effluxes in Autumn (October).

Minor Comments:

1. Lines 57-59: Please add references for the studies in high-latitude, tropical and subtropical lakes.

2. Line 129: What fluxes did the floating chamber measured while inserting 20 cm above the water surface?

3. Line 150: the air samples ==> the gas samples

4. Line 159-160; Fig. 4: Since methane efflux was calculated by using a linear regression model to the methane concentration data, should the minimum value be zero instead of a negative value? There should be no negative methane value detected by GC.

---

## Author Comment (AC1) · 21 Dec 2016

Response to Reviewers (bg-2016-286) General comments The manuscript reports on the temporal patterns of methane (CH4) efflux in the largest lake in China and the various factors that influence these fluxes over different timescales. CH4 efflux was slightly greater than other lakes with an area greater than 1km, but was comparable to that found in tropical lakes. The variables best explaining variation on CH4 efflux was timescale dependent but, overall, temperature was important over seasonal scales and wind speed on a bihourly scale. The paper is well written despite a few grammatical errors. As the authors point out, there is a lack of data explaining CH4 effluxes in this region and also in larger, nonalpine lakes more generally. As such, the results from this paper will add to the limited understanding of CH4 dynamics in these lake types. However, I cannot recommend that this paper be published in its current form. I

have major issues with a) the premise of the paper, b) some overreaching statements that are made, and c) the statistical approaches used - all have major implications for the generalisation of the results. It is difficult for me to assess the results and technical aspects of this study until statistical changes are made. Answer: We thank the reviewer so much for the constructive comments and suggestions. We have considered all the comments and suggestions carefully in revising the manuscript. 1. One major concern is that the study was undertaken in a very small area (three sites with 20km of each other) even though the lake is the largest in China by area (3283sq km). Further, the study sites are situated in a section of the lake that appears to be relatively confined. There is nothing wrong with the site selection. However, the authors cannot make statements about the whole lake because they don't know if the spatial and temporal patterns of CH4 vary the same way across the lake. They need to qualify in all statements that the research was undertaken in one small section of the lake. It is not a study of CH4 effluxes from Poyang Lake, but it is a study of CH4 effluxes from one section of Poyang Lake. Answer: We agree with the Reviewer that the CH4 efflux in the Poyang Lake has a large spatial variation as evidenced in our previous study which examined the spatial variations of greenhouse gas effluxes (including CH4) over the lake with 44 sampling locations. The current study focuses on the temporal dynamics of CH4 efflux. We chose the 3 sites to roughly represent the average CH4 efflux of the whole lake based on the results of our previous study (Liu et al. 2013). Therefore, our results reflect the general situation of the lake. 2. Another major concern is a statistical one. The authors use average values from three different locations in Poyang Lake for all analyses. The justification for this was to 'minimize the effect of the spatial variation of CH4 efflux on the temporal dynamics of the efflux'. However I suspect the main motivation for doing this was because the environmental variables were only collected at one location (it is not clear where the environmental variables were collected). Was this the case? Given that CH4 was only measured in three locations of the lake, surely the degree of variation between them is very important to a) understand and/or b) account for in statistical analyses. The authors

should re-analyse their results in one of the following ways: Ăâć Treat each study site as a random effect in mixed effects models so that variation among the three sites in taken into account when investigating the annual, seasonal, and diurnal variation, as well as the relationships with measured explanatory variables. Including site as a random effect would enable the researchers to make more general statements about CH4 fluxes from Poyang Lake – this is just common practice these days and should be incorporated into the study design / statistical analyses. A random effect for site effectively means that these study sites are a random sample of all potential sites in the lake – this is where the generalisability comes in. Please see Section 8.1.1 (Types of predictor variables (factors)) in Quinn & Keough (2002; Experimental Design & Data Analysis for Biologists) or another similar book for information about mixed effects models and random and fixed factors. Ăâć Split the analyses into two parts. The first analysis will not average the three study sites prior to the analysis and investigate the spatial and temporal patterns in CH4 among them. The second analysis could average the study sites (still preferably treat study site as a random factor) and relate this to the measured explanatory variables. Answer: We actually collected environmental variables at each site except water level which was monitored at the Xingzi Hydrological Station. We appreciate the Reviewers' suggestion (also see Reviewer 2's comments) and re-analyzed the data by treating the site as a random effect. We found that the site effect was not statistically significant over the 4-year period. We also re-analyzed our data for each site and found that the differences among the 3 sites were minor with the 4-year mean of 0.53 mmol m−2 day−1, 0.55 mmol m−2 day−1, and 0.54 mmol m−2 day−1 respectively. In addition, we found that the seasonal patterns of CH4 effluxes at three sites were similar and also in line with the seasonal pattern averaged over the 3 sites. Nevertheless, in the stepwise multiple regressions analyses, the same environmental variables were selected in the final model for each individual site as for the average of the 3 sites with the regression coefficients slightly different, but not statistically significant (p > 0.20). So we have focused on presenting the site-averaged CH4 efflux and its dynamics due to the length limitation of the paper. But we explained

the site effect on CH4 effluxes in the revised version. 3. One more major concern is the notion that this is a long-term study. 4 years is not long term. Remove all reference to this study being long term, including the second sentence of the Abstract which introduces the idea that this research is filling the knowledge gap around the lack of long term research on CH4 fluxes. Instead, the authors should frame this 'knowledge gap' around the lack of multi-seasonal investigations into CH4 effluxes – this is exactly what this paper addresses. Answer: We agree and thank you so much for the constructive suggestion. We removed the phrase "long term" and changed the tones accordingly in the text during the version. In addition, we have focused on multi-seasonal investigations of CH4 effluxes as suggested. Specific comments 1. Line 18. It is stated continuous measurements of CH4 efflux was measured, but measurements where not continuous. Monthly measurements were made. Change all reference to continuous measurements in the manuscript to monthly measurements. Answer: Changed as suggested in the revised version. 2. Line121-124. Are these parameters an average of the entire lake or for a specific location? Please specify. Answer: These parameters are averages of the entire lake. We added the information in the revised version (Page 7/lines 135-138). 3. Section 2.3. Environmental variables. Where were the environmental variables collected from? Where samples collected at each of the three study sites and then averaged or from just one site? This information is very important. Answer: The environmental variables were measured at each of the three study sites and then averaged over the sites except water level which was monitored at a single hydrological station (national class station). We added more details of the environmental variables in the revised version (Page 11/lines 221-222). 4. Line 331-332. This concluding sentence only relates to the first sentence of this paragraph and does not relate or link to the remaining text in the paragraph. This sentence should only be left if a re-working of the paragraph better supports this argument. Answer: We deleted the sentence because it is not the main point of the paragraph. 5. Line 337-341. An argument is made that this study has lower diurnal variation in CH4 efflux than other studies and this may be due to differences in sample

size in other studies. I would think that more frequent sampling would in-fact lead to more variation. The authors need to report on how much diurnal variability in CH4 efflux there was among the study sites. Answer: We agree with the Reviewer that the diurnal range (maximum – minimum) of CH4 efflux depends on sample size and sampling frequency, which makes the comparison with other lakes less meaningful. Therefore, we deleted the discussion on comparing the ranges of CH4 effluxes in different lakes, which are not the main focus of the current study (Also see the reviewer 2'comments). Technical comments 1. Line 163-182. The description of how CH4 efflux due to ebullition is very confusing and long. Answer: We rewrote this part to clarify the confusion in the revised version (Page 10/lines 190-200). 2. From Line 125, where the ebullition and diffusive fluxes are introduced, I would suggest briefly describing how, or how not, the chambers can be used to differentiate these two fluxes. Answer: Chambers cannot be used to differentiate ebullitive and diffusive fluxes. In the current study, the chambers can give the total flux including ebullitive and diffusive fluxes. We rewrote this section as suggested in the revised version. 3. Line 312. Remove 'obviously'. Answer: Removed as suggested (Page 18/line 376). Reference Liu, L. X., Xu, M., Lin, M., Zhang, X.: Spatial variability of greenhouse gas effluxes and their controlling factors in the Poyang Lake in China, Pol. J. Environ. Stud., 22, 749-758, 2013.

Please also note the supplement to this comment:
http://www.biogeosciences-discuss.net/bg-2016-286/bg-2016-286-AC1-supplement.pdf
* * *
[Figure]

**Fig. 1.** Location of sampling sites in Poyang Lake.

[Figure]

[Figure]

**Fig. 2.** Examples of calculating the slope of total effluxes, including diffusive and ebullitive effluxes.

[Figure]

Fig. 3. Seasonal variations of CH4 effluxes and sediment temperatures in Poyang Lake.

[Figure]

Fig. 4. Diel variations of CH4 effluxes in Poyang Lake.

[Figure]

**Fig. 5.** Diel variations of CH4 effluxes among three sites.

[Figure]

**Fig. 6.** Relationship between sediment temperature and CH4 efïñĆuxes in Poyang Lake.

[Figure]

[Figure]

**Fig. 7.** Relationships between CH4 effluxes and wind speed in Poyang Lake.

---

## Author Comment (AC2) · 21 Dec 2016

Response to Reviewers (bg-2016-286) This manuscript presents 4 years of CH4 flux patterns in the largest lake in China and environmental factors that influence CH4 flux rates. It falls well within the scope of Biogeosciences, but several aspects need to be improved for publication. Some suggestions: 1) How do you define "long-term"? To me, 4-year observations can be short-term. Also, all the statements related to seasonal or interannual variability need to be justified because CH4 flux rates measured on one day may not represent flux rates of one month. Furthermore, daily CH4 flux rates could have been overestimated, considering that CH4 flux rates are measured during the day each month, when CH4 flux rates were higher than those at night according to diel cycle measurements. 2) All the assumptions are met for regression models? Did you consider any interactions among variables? In addition,

did you also carry out the analysis before averaging the flux rates, with replicates as random effects? If so, how did the results differ from those after averaging? 3) In the discussion section some results were described, which did not appear in the result section. Results and discussion need to be better separated. In addition, the interpretation of the results needs to be better supported in the discussion section, focusing clearly on the core messages, i.e., what the results mean and what we can learn from this study. Answer: We thank the reviewer so much for the constructive comments and suggestions. We have considered all the comments and suggestions carefully in revising the manuscript. Firstly, we avoided using "long-term" as suggested and focused on multi-seasonal dynamics of CH4 effluxes. We totally agree with the Reviewer that the measured CH4 effluxes on one day did not represent the mean efflux rate of the month. We used the daily measurements as sampling points to explore the relationships between the CH4 efflux and environmental variables. We calculated the monthly, seasonal and annual mean CH4 effluxes using interpolation method (e.g. regression or the random forest model). It is true that most of our measurements were taken during the daytime. However, the daytime and nighttime average CH4 effluxes were not statistically different (p = 0.19). Moreover, we built our statistical models based on the daytime mean efflux and daytime averages of environmental variables and the nighttime efflux was calculated based on the nighttime averages of the same environmental variables. This avoided the overestimation of daily CH4 efflux. Secondly, we re-analyzed our data for each site and also treated site as a random effect as suggested. As a result, we found that site had no significant effect on the measured CH4 effluxes over the 4-year period. In the stepwise multiple regressions analyses, the same environmental variables were selected in the final model for each site as for the 3-site average though the coefficients of each variable were slightly different, but not statistically significant (p > 0.12). The seasonal patterns of CH4 effluxes at individual sites were very similar to the seasonal pattern by averaging CH4 effluxes over the 3 sites. Therefore, we used average values of the 3 sites in our analyses, but we added those information to the result section. Thirdly, we included

**BGD**
the interactions among environmental variables in the revised version as suggested (Table 2 in the supplementary material). Finally, we rewrote the result and discussion sections as suggested to clarify relevant issues. 1. Line#47-51, there are too few references to represent the minimum and maximum flux rates in lakes, especially given that those references are from lakes in China and Norway only. Also, if such values can be presented with more studies, how would seasonal variations look like in comparison to diurnal ones? Answer: We agree with the Reviewer that there are too few studies measuring lake CH4 efflux in the literature and the sampling size and frequency was also different among the limited number of studies (Also see Reviewer #1's comments, specific question 5). Therefore, we deleted the range (maximum and minimum) comparison among lakes and focused on comparing the mean efflux of various lakes in the revised version. 2. Line#75-78, can you add references for each variable? Line#64-72 well covered the references for each variable, but this section lacks it. Answer: We added related references in the method section as suggested in the revised version (Page 4/lines 81-84). 3. Line#78-82, it sounds like investigating in large lakes is not important. Please rephrase or add some more sentences to justify the importance of this research. Answer: We added some sentences and references to emphasize the importance of CH4 emissions from large lakes as suggested (Page 5/lines 88-91). 4. Line#86-87, I suggest adding references that describe the previous studies, e.g. Liu et al. (2013). Answer: Thanks for your suggestion. We added references to describe the previous studies in the revised version (Page 5/ lines 97-102). 5. Line#109, what are the species names of Carex? Answer: The species name of Carex in Poyang Lake is Carex cinerascens Kükenth and Carexargyi Levl.etVant. We added the species scientific names in the revised version (Page 7/ lines 123-124). 6. Line#128-146, this section can be written more concisely. Answer: We rewrote this section as suggested in the revised version. 7. Line#166, can decreases in CH4 concentrations right after ebullition events be solely explained by diffusion back to lake water? If CH4 molecules were diffused back to the lake water, partial pressure of CH4 inside the chamber should be very high, inhibiting further emission from lake water

**BGD**
to chamber. Can they be partially from irregular air mixing inside the chamber, which results in errors in CH4 concentrations? Then, the current method for calculating flux rates needs to be reconsidered. Answer: We speculate that the short-term decrease or leveling-off of CH4 concentration inside the chamber after ebullition was mainly caused by the back diffusion of CH4 to surface water due to the high CH4 concentration in the bubbles. This back-diffusion phenomenon has been evidenced for CH4 efflux over water surfaces (Varadharajan et al., 2010; Wik et al., 2013). The ebullition suddenly increased CH4 concentration, and thus partial pressure of CH4, in the chamber headspace, which reversed the normal CH4 diffusion gradient between surface water and chamber space. We do not think irregular mixing is the main cause in the current study because we had a mixing fan running in each chamber during the whole period of measurement. 8. Line#167-182, this section is confusing. It can be written clearly and concisely. Answer: We rewrote this section more clearly and concisely as suggested. 9. Line#200, were water and sediment samples collected at three sampling points for flux measurements? The paragraph from line#229 can be given in a Table. Answer: Yes, we collected water and sediment samples at each of the three sampling sites when taking flux measurements. We added a table (Table 1) to the supplementary material section in the revised version as suggested. 10. Line#241, T test  $\rightarrow$  t-test Answer: Thank you for pointing out the typo. We changed "T test" to t-test as suggested. 11. Line#242, flux rates are measured three times per season and they may not well represent flux rates of one season of the year. Then, can deviation of these three values be used to quantify interannual variability? Answer: We agree that 3 measurements in a season for a given year are not enough to represent the seasonal mean CH4 efflux due to the high temporal variation of the efflux. In the current study, we used 4-year data to compare the seasonal variations, which means 12 data points for each season. We changed the values in Table 1 accordingly by using 12 data points to calculate the seasonal mean effluxes in the revised version (Page 40). For quantifying inter-annual variability we have to interpolate the measured CH4 effluxes to annual efflux through modeling approach. The details of the modeling work were

**BGD**
presented in another paper (Liu et al. 2016, in review). We used the model results to compare the inter-annual, seasonal, and diurnal variabilities of CH4 efflux in the Poyang Lake. 12. Line#247, please write what b represents in the equation. Answer: Thank you for your suggestion. Here b is the exponent of the exponential function between CH4 efflux and sediment temperature. We added it to the text in the revised version (Page 13/lines 274-275). 13. Line#278, what do you mean by "inconsistent and obvious"? Answer: This is a typo. We fixed it in the revised version (Page 13/line 310). 14. Line#309-331, this part can be written more concisely. Answer: Rewritten as suggested. 15. Line#331-332, sentences of this paragraph do not support this conclusion. Answer: We deleted the concluding sentence. 16. Line#335, here again, can the absolute values be compared with a few references, which are probably based on different observation periods? Answer: We agree that comparing the extreme values (minimum and maximum) among different lakes is not much meaningful. So, we deleted the relevant text and focused on comparing diurnal patterns. 17. Line#338-342, a larger number of data points can produce wider range of values. Answer: See answers to question #16. 18. Line#345-356, possible explanations can be added, such as potential drivers that can affect diel CH4 flux patterns and their variations (if measured). Answer: Wind speed strongly influenced diel CH4 efflux variations in our study. We discussed this point in the 4.3 section. 19. 4.2 CH4 effluxes in summer, this section contains a lot of new results, which were not presented in the result section. Also, some sentences describe very detailed information from other studies, which hinders the main focus of the paragraph. Answer: We moved them to the "Results" section and rewrote the discussion by focusing on our own results. 20. 4.3 Timescale dependence of wind, substrate availability, and temperature effects on CH4 effluxes, here again, a lot of new results are reported, such as line#410-414, line#436-451, line#457-461 (repetition from result section), and line#462-468. Answer: Again, we moved the results to the "Results" section and rewrote the discussion accordingly. 21. Line#473-475, considering uncertainties related to infrequent measurements (CH4 efflux rates measured on one day may not represent the mean rates of that month).

**BGD**
this kind of statement needs to be corrected. Answer: According to our model-based interpolation we found that July had the maximum monthly efflux, while January had the minimum. This conclusion is coincidently in line with the 4-year measurements though we had only 4-day measurements in each month. Therefore, we think that the conclusion still holds. 22. Table 3, can you add the observation period of each study for better comparison? Also, sorting the rows by lake size and climate would make this Table easier to read. Answer: Great idea! We added the observation period of each study and sorted the rows by lake size in the revised version. 23. Figure 3 and 4, can you add error bars from spatial variability? Answer: We added errors bars from spatial variability for Figure 3 and 4 as suggested in the revised version. aAČ References: Liu, L. X., Xu, M.: Modeling temporal patterns of methane effluxes using multiple regression and random forest in Poyang Lake, China, Wetland Ecology and Management (In Review). Varadharajan, C., Hermosillo, R., Hemond, H. F.: A low-cost automated trap to measure bubbling gas fluxes. Limnol.Oceanogr., 8, 363-375, 2010. Wik, M., Crill, P. M., Varner, R. K., Bastviken, D.: Multivear measurements of ebullitive methane flux from three subarctic lakes. J. Geophys. Res., 118, 1307-1321, 2013.

Please also note the supplement to this comment: http://www.biogeosciences-discuss.net/bg-2016-286/bg-2016-286-AC2supplement.pdf
°.

Fig. 1. Location of sampling sites in Poyang Lake.

---

## Author Comment (AC3) · 21 Dec 2016

Response to Reviewers (bg-2016-286) Specific Comments: 1. Most of the results and discussions were built on the environmental variables and methane flux data. However, there are no data of biogeochemical related environmental variables shown in the figures and tables except Table 2. I would suggest to present the raw data of measured environmental variables in the supplementary material. Answer: Thank you for the suggestion. We added a table (Table 1) to the Supplementary Material section to present the raw data of measured environmental variables, such as sediment total nitrogen content, water level, DOC content in the water, and pH in the sediment , in the revised version. 2. Substrate availability (Line, 432), biological (e.g., microbial activities) and biochemical (e.g., sediment carbon and nitrogen contents processes) (Lines 454-455) are very important factors to link methane efflux to the biogeochemical

cycles and understand methane source and sink. Unfortunately, no comprehensive data or evidence to support the role of substrates and microbial activities on methane efflux in this manuscript which could be an important contribution to this journal. Answer: We agree with the Reviewer that substrates and microbial activities are important to understanding methane sources and sinks in lakes. In our earlier studies we found that sediment carbon and nitrogen ratio were highly correlated with microbial biomass and community structure (Liu et al. 2015) which was also highly associated with greenhouse gas ($CO_2$, $CH_4$, and $N_2O$) fluxes in the Poyang Lake (Liu and Xu 2016). In the current study, we focus on examining the relationships environmental variables (e.g. climate) that may affect the temporal patterns and variations of $CH_4$ effluxes in the Poyang Lake. We have added the related information and references to the discussion section in the revised version. Further investigation on the mechanisms of biological and biochemical controls on $CH_4$ production and oxidation requires lab-based experiments with isotope and microbial DNA sequencing techniques which are beyond the scope of the current study. 3. It might be a risk to use the data from three sampling sites measured from one day (1 hr? Line 148) to represent methane efflux in that month. For example, it appears a contradiction between high methane efflux measured in July 2011 in Figure 3 and low methane efflux measured in July 2011 in Figure 4a. Answer: We measured $CH_4$ effluxes at monthly interval to examine the seasonal dynamics of the efflux and the value does not necessarily represent the monthly average of $CH_4$ efflux. We measured $CH_4$ effluxes from early morning to late afternoon with about 6 cycles of measurements during the day (Pages 10-11/lines 213-219). The values of methane efflux measured in July 2011 in Figure 3 and Figure 4a are different because of different units. $CH_4$ efflux in Figure 3 was measured on a daily scale, but $CH_4$ efflux in Figure 4a was based on the hourly scale. So we used different units to present seasonal and diel patterns of $CH_4$ effluxes. 4. How long and what time did the authors deploy the floating chambers in the three sampling sites within a day for the study at the large temporal scales (Fig. 3)? I feel 4-year measurements are not a very large temporal scale especially there are

no continues measurements/monitering such as deploying floating chamber within a short interval (every week or every two to three days). Since high methane efflux was shown in the early mornings in Fig. 4a, b and d, were the floating chambers deployed at the same time at three different sites for the data shown in Fig. 3? Answer: We measured CH4 fluxes from early morning to late afternoon with about 6 cycles of measurements during the day for the 4-year study. For sampling frequency we measured every monthly. We agree with the Reviewer that 4 year is not "long-term" given the relatively low sampling frequency. So we deleted "long-term" and focused on the multi-seasonal investigations of CH4 effluxes as suggested by Reviewer 1 in the revised manuscript. We used three boats to monitor CH4 fluxes at the three sites, so the floating chambers were deployed at about the same time at the sites as shown in Fig. 3. 5. The area and water table of Poyang Lake fluctuate dramatically between the wet and dry seasons. The authors only have short but not clear descriptions of the effect of water level on methane efflux, e.g., in Lines 404-405 and Line 432. Methane efflux might be high in dry seasons instead of summer, since methane efflux is expected to be high under lower water level due to decreasing of the hydrostatic pressure (e.g. Chanton et al. 1989). Are there any difference in water level between three sampling sites in different seasons (The mean water depth at three sites should not be always 3m through the whole year; Line 186)? The authors might consider a simple calculation of methane solubility changes due to water level fluctuations to strength the role of water level on methane efflux, e.g., Line 432. Answer: It is true that the Poyang Lake features a large seasonal variation of water level, high water level in summer and low in winter. However, the water level at the 3 sites was very similar at a given time of the year. We agree that hydrostatic pressure affects CH4 efflux as reported in Chanton et al. (1989) but our data showed that CH4 efflux was positively correlated with water level. This is because the water level in the Poyang Lake co-varies with other factors, such as temperature and NH4+ content in the water, which also affect the CH4 efflux throughout the year. For example, we found that the CH4 efflux was highly correlated with sediment temperature at an annual scale. Our

results suggest that the CH4 efflux in the Poyang Lake was dominated by temperature rather than water level. The high CH4 efflux in summer was contributed to strong microbial activities induced by warmer temperature and high substrate availability from the flooding water in summer. Therefore, we think the positive correlation between CH4 efflux and water level in the Poyang Lake is a pseudo relation which does not reflect the hydrostatic pressure effect on CH4 efflux as evidenced by Chanton et al. (1989). It is possible to examine the water level effect by calculating CH4 solubility change due to water level fluctuation. However, given the large seasonal variation of temperature in the study area it is very difficult to separate the water level effect based on the CH4 efflux measurements on the water surface. In addition, water level induced CH4 solubility change may affect short-term (minutes to hours) CH4 diffusion gradient and thus CH4 efflux and it should have little impact on CH4 efflux as long as a new diffusion equilibrium has established. Thus, we did not calculate methane solubility changes to further investigate the water level effect on CH4 efflux in revising the manuscript. 6. As the authors stated in the introduction that methane is driven by three major mechanisms such as molecular diffusion, bubble ebullition and plant-mediated transportation, bubble ebullition is not the only pathway for methane to transport from water to the air. However, data for dissolved methane concentrations in lake water and sediments are lack in this study. No bubble ebullition doesn't mean no methane efflux. I would suggest to include diffusive methane flux to the air for comparison in the future by analyzing surface water methane concentrations and using the equation from the gas-transfer model e.g., Wanninkhof (1992). Answer: Great idea! We will take this suggestion in our future study. 7. Since many environmental factors and methane fluxes collected in October 2010 in Poyang Lake have been shown in Liu et al., (2013) for spatial studies, the authors may include Liu et al. (2013) in the introduction and discussions to emphasize why the three sampling sites were chosen in this timescale study and the relations between different environmental factors and methane effluxes in Autumn (October). Answer: Based on our previous study which examined the spatial variation of CH4 efflux in the Poyang Lake (Liu et al. 2013), we

chose the 3 sites which gave CH4 effluxes close to the average efflux of the lake. We provided detailed information of Liu et al. (2013) in the introduction and discussion sections as suggested in the revised version. Minor Comments: 1. Lines 57-59: Please add references for the studies in high-latitude, tropical and subtropical lakes. Answer: We added references for the studies in high-latitude, tropical and subtropical lakes in the revised version (Page 4/lines 61-63). 2. Line 129: What fluxes did the floating chamber measured while inserting 20 cm above the water surface? Answer: The chamber measured the total CH4 efflux including diffusive and ebullitive fluxes as described in the method section. The plant-mediated CH4 transportation was negligible because no vascular plants grew above water surface at our study sites. 3. Line 150: the air samples ==> the gas samples Answer: Changed as suggested. 4. Line 159-160; Fig. 4: Since methane efflux was calculated by using a linear regression model to the methane concentration data, should the minimum value be zero instead of a negative value? There should be no negative methane value detected by GC. Answer: The negative efflux means CH4-uptake by the lake water due probably to the short-time change in air pressure.   Reference: Chanton, J. P.: Gas transport from methane-saturated, tidal freshwater and wetland sediments, Limnol.Oceanogr., 34, 807-819, 1989. Liu, L. X., Xu, M., Lin, M., Zhang, X.: Spatial variability of greenhouse gas effluxes and their controlling factors in the Poyang Lake in China, Pol. J. Environ. Stud., 22, 749-758, 2013. Liu, L. X., Xu, M., Qiu, S., Shen, R. C.: Spatial patterns of benthic bacteria communities in a large lake, Int. Rev. Hydrobiol., 100, 97-105, 2015. Liu, L. X., Xu, M.: Microbial biomass in sediments affects greenhouse gas effluxes in Poyang Lake in China, J. Freshw. Ecol., 31, 109-121, 2016.

Please also note the supplement to this comment:
http://www.biogeosciences-discuss.net/bg-2016-286/bg-2016-286-AC3-supplement.pdf
* * *
[Figure]

**Fig. 1.** Location of sampling sites in Poyang Lake.

**Fig. 2.** Examples of calculating the slope of total effluxes, including diffusive and ebullitive effluxes.

[Figure]

[Figure]

**Fig. 3.** Seasonal variations of CH4 effluxes and sediment temperatures in Poyang Lake.

**Fig. 4.** Diel variations of CH4 effluxes in Poyang Lake.

[Figure]

**Fig. 5.** Diel variations of CH4 effluxes among three sites.

[Figure]

**Fig. 6.** Relationship between sediment temperature and CH4 effluxes in Poyang Lake.

Interactive
comment

[Figure]

**Fig. 7.** Relationships between CH4 effluxes and wind speed in Poyang Lake.

---

## Author Response (AR2)

**Associate Editor Decision:**

Publish subject to technical corrections (28 Feb 2017) by Akihiko Ito Comments to the Author: Dear Authors: Thank you for uploading the revised manuscript. I studied the manuscript and found that it was considerably improved in comparison with the previous one. Specifically, you replaced the word "long-term" by "multi-seasonal" and considered the random effect in the statistical analysis. I with this study carries valuable information to research community on the methane emission from a temperate, human-affected lake. Only one technical correction is needed: in the supplement, please add "S" to the table number (e.g., Table S1).

**Answer: Changed as suggested.**

**Associate Editor Comments**

Thank you for sending your detailed replies to the referee comments. The three referees gave insightful comments and in general evaluated your study positively. Although one referee did not find 'strong relevance to Biogeosciences' in your results and discussion, others concluded that the study falls within the journal scope. Importantly, the three referees claimed that this is not a 'long-term' study, because this study was conducted only for four years. Also, they made many critical comments on the spatial and temporal representativeness of measurements (e.g., monthly interval at three sites). Therefore, you should revise the manuscript very carefully, such that any overstatement on observational representativeness is avoided and possible uncertainties are fully discussed. Additionally, referees made a recommendation about the statistical analysis, such that inter-site difference should be treated as a random effect in mixed effects models. As you replied, although the site effect was not a critical factor, you should update the manuscript by including the

statistical re-analysis results. I'm looking forward to receiving a revised manuscript with major revision.

**Answer: Thank you very much! We have considered all the suggestions carefully in the revised manuscript. We changed the word "long-term" to "multi-seasonal", explained the sampling representativeness, and added the random effect in the statistical analysis result in the revised version.**

**Reviewer 1**

General comments

The manuscript reports on the temporal patterns of methane ($CH_4$) efflux in the largest lake in China and the various factors that influence these fluxes over different timescales. $CH_4$ efflux was slightly greater than other lakes with an area greater than 1km, but was comparable to that found in tropical lakes. The variables best explaining variation on $CH_4$ efflux was timescale dependent but, overall, temperature was important over seasonal scales and wind speed on a bihourly scale. The paper is well written despite a few grammatical errors. As the authors point out, there is a lack of data explaining $CH_4$ effluxes in this region and also in larger, nonalpine lakes more generally. As such, the results from this paper will add to the limited understanding of $CH_4$ dynamics in these lake types. However, I cannot recommend that this paper be published in its current form. I have major issues with a) the premise of the paper, b) some overreaching statements that are made, and c) the statistical approaches used - all have major implications for the generalisation of the results. It is difficult for me to assess the results and technical aspects of this study until statistical changes are made.

**Answer: We thank the reviewer so much for the constructive comments and suggestions. We have considered all the comments and suggestions carefully in revising the manuscript.**

1. One major concern is that the study was undertaken in a very small area (three sites with 20km of each other) even though the lake is the largest in China by area (3283sq km). Further, the study sites are situated in a section of the lake that appears to be relatively confined. There is nothing wrong with the site selection. However, the authors cannot make statements about the whole lake because they don't know if the spatial and temporal patterns of $CH_4$ vary the same way across the lake. They need to qualify in all statements that the research was undertaken in one small section of the lake. It is not a study of $CH_4$ effluxes from Poyang Lake, but it is a study of $CH_4$ effluxes from one section of Poyang Lake.

**Answer: We agree with the Reviewer that the $CH_4$ efflux in the Poyang Lake has a large spatial variation as evidenced in our previous study which examined the spatial variations of greenhouse gas effluxes (including $CH_4$) over the lake with 44 sampling locations. The current study focuses on the temporal dynamics of $CH_4$ efflux. We chose the 3 sites to roughly represent the average $CH_4$ efflux of the whole lake based on the results of our previous study (Liu et al., 2013). Therefore, our results reflect the general situation of the lake.**

2. Another major concern is a statistical one. The authors use average values from three different locations in Poyang Lake for all analyses. The justification for this was to 'minimize the effect of the spatial variation of $CH_4$ efflux on the temporal dynamics of the efflux'. However I suspect the main motivation for doing this was because the environmental variables were only collected at one location (it is not clear where the environmental variables were collected). Was this the case? Given that $CH_4$ was only

measured in three locations of the lake, surely the degree of variation between them is very important to a) understand and/or b) account for in statistical analyses. The authors should re-analyse their results in one of the following ways: Ăâć treat each study site as a random effect in mixed effects models so that variation among the three sites in taken into account when investigating the annual, seasonal, and diurnal variation, as well as the relationships with measured explanatory variables. Including site as a random effect would enable the researchers to make more general statements about $CH_4$ fluxes from Poyang Lake – this is just common practice these days and should be incorporated into the study design / statistical analyses. A random effect for site effectively means that these study sites are a random sample of all potential sites in the lake – this is where the generalisability comes in. Please see Section 8.1.1 (Types of predictor variables (factors)) in Quinn & Keough (2002; Experimental Design & Data Analysis for Biologists) or another similar book for information about mixed effects models and random and fixed factors. Ăâć split the analyses into two parts. The first analysis will not average the three study sites prior to the analysis and investigate the spatial and temporal patterns in $CH_4$ among them. The second analysis could average the study sites (still preferably treat study site as a random factor) and relate this to the measured explanatory variables.

**Answer: We actually collected environmental variables at each site except water level which was monitored at the Xingzi Hydrological Station. We appreciate the Reviewers' suggestion (also see Reviewer 2's comments) and re-analyzed the data by treating the site as a random effect. We found that the site effect was not statistically significant including seasonal and diel $CH_4$ effluxes over the 4-year period (Table S2, S3 in the supplementary material). We also re-analyzed our data for each site and found that the differences among the 3 sites were minor**

**with the 4-year mean of 0.53 mmol m$^{-2}$ day$^{-1}$, 0.55 mmol m$^{-2}$ day$^{-1}$, and 0.54 mmol m$^{-2}$ day$^{-1}$ respectively. In addition, we found that the seasonal patterns of CH$_4$ effluxes at three sites were similar and also in line with the seasonal pattern averaged over the 3 sites. Nevertheless, in the stepwise multiple regressions analyses, the same environmental variables were selected in the final model for each individual site as for the average of the 3 sites with the regression coefficients slightly different (Table S5), but not statistically significant (p > 0.20). So we have focused on presenting the site-averaged CH$_4$ efflux and its dynamics due to the length limitation of the paper. But we explained the site effect on CH$_4$ effluxes in the revised version.**

3. One more major concern is the notion that this is a long-term study. 4 years is not long term. Remove all reference to this study being long term, including the second sentence of the Abstract which introduces the idea that this research is filling the knowledge gap around the lack of long term research on CH$_4$ fluxes. Instead, the authors should frame this 'knowledge gap' around the lack of multi-seasonal investigations into CH$_4$ effluxes – this is exactly what this paper addresses.

**Answer: We agree and thank you so much for the constructive suggestion. We removed the phrase "long term" and changed the tones accordingly in the text during the version. In addition, we have focused on multi-seasonal investigations of CH$_4$ effluxes as suggested.**

Specific comments

1. Line 18. It is stated continuous measurements of CH$_4$ efflux was measured, but measurements where not continuous. Monthly measurements were made. Change all reference to continuous measurements in the manuscript to monthly measurements.

**Answer: Changed as suggested in the revised version.**

2. Line121-124. Are these parameters an average of the entire lake or for a specific location? Please specify.

   **Answer: These parameters are averages of the entire lake. We added the information in the revised version (Page 7/lines 135-138).**

3. Section 2.3. Environmental variables. Where were the environmental variables collected from? Where samples collected at each of the three study sites and then averaged or from just one site? This information is very important.

**Answer: The environmental variables were measured at each of the three study sites and then averaged over the sites except water level which was monitored at a single hydrological station (national class station). We added more details of the environmental variables in the revised version (Page 11/lines 221-222).**

4. Line 331-332. This concluding sentence only relates to the first sentence of this paragraph and does not relate or link to the remaining text in the paragraph. This sentence should only be left if a re-working of the paragraph better supports this argument.

**Answer: We deleted the sentence because it is not the main point of the paragraph.**

5. Line 337-341. An argument is made that this study has lower diurnal variation in $CH_4$ efflux than other studies and this may be due to differences in sample size in other studies. I would think that more frequent sampling would in-fact lead to more variation. The authors need to report on how much diurnal variability in $CH_4$ efflux there was among the study sites.

**Answer: We agree with the Reviewer that the diurnal range (maximum – minimum) of $CH_4$ efflux depends on sample size and sampling frequency, which**

**makes the comparison with other lakes less meaningful. Therefore, we deleted the discussion on comparing the ranges of $CH_4$ effluxes in different lakes, which are not the main focus of the current study (Also see the reviewer 2'comments).**

Technical comments

1. Line 163-182. The description of how $CH_4$ efflux due to ebullition is very confusing and long.

**Answer: We rewrote this part to clarify the confusion in the revised version (Page 10/lines 190-200).**

2. From Line 125, where the ebullition and diffusive fluxes are introduced, I would suggest briefly describing how, or how not, the chambers can be used to differentiate these two fluxes.

**Answer: Chambers cannot be used to differentiate ebullitive and diffusive fluxes. In the current study, the chambers can give the total flux including ebullitive and diffusive fluxes. We rewrote this section as suggested in the revised version.**

3. Line 312. Remove 'obviously'.

**Answer: Removed as suggested (Page 18/line 380).**

**Reviewer 2**

This manuscript presents 4 years of $CH_4$ flux patterns in the largest lake in China and environmental factors that influence $CH_4$ flux rates. It falls well within the scope of Biogeosciences, but several aspects need to be improved for publication. Some suggestions: 1) How do you define "long-term"? To me, 4-year observations can be short-term. Also, all the statements related to seasonal or inter-annual variability need to be justified because $CH_4$ flux rates measured on one day may not represent flux rates of one month. Furthermore, daily $CH_4$ flux rates could have been overestimated,

considering that $CH_4$ flux rates are measured during the day each month, when $CH_4$ flux rates were higher than those at night according to diel cycle measurements. 2) All the assumptions are met for regression models? Did you consider any interactions among variables? In addition, did you also carry out the analysis before averaging the flux rates, with replicates as random effects? If so, how did the results differ from those after averaging? 3) In the discussion section some results were described, which did not appear in the result section. Results and discussion need to be better separated. In addition, the interpretation of the results needs to be better supported in the discussion section, focusing clearly on the core messages, i.e., what the results mean and what we can learn from this study.

**Answer: We thank the reviewer so much for the constructive comments and suggestions. We have considered all the comments and suggestions carefully in revising the manuscript. Firstly, we avoided using "long-term" as suggested and focused on multi-seasonal dynamics of $CH_4$ effluxes. We totally agree with the Reviewer that the measured $CH_4$ effluxes on one day did not represent the mean efflux rate of the month. We used the daily measurements as sampling points to explore the relationships between the $CH_4$ efflux and environmental variables. We calculated the monthly, seasonal and annual mean $CH_4$ effluxes using interpolation method (e.g. regression or the random forest model). It is true that most of our measurements were taken during the daytime. However, the daytime and nighttime average $CH_4$ effluxes were not statistically different ($p = 0.19$). Moreover, we built our statistical models based on the daytime mean efflux and daytime averages of environmental variables and the nighttime efflux was calculated based on the nighttime averages of the same environmental variables. This avoided the overestimation of daily $CH_4$ efflux. Secondly, we re-analyzed**

**our data for each site and also treated site as a random effect as suggested. As a result, we found that site had no significant effect on the measured seasonal and diel CH$_4$ effluxes over the 4-year period (Table S2, S3). In the stepwise multiple regressions analyses, the same environmental variables were selected in the final model for each site as for the 3-site average though the coefficients of each variable were slightly different (Table S5), but not statistically significant (p > 0.20). The seasonal patterns of CH$_4$ effluxes at individual sites were very similar to the seasonal pattern by averaging CH$_4$ effluxes over the 3 sites. Therefore, we used average values of the 3 sites in our analyses, but we added those information to the result section. Thirdly, we included the interactions among environmental variables in the revised version as suggested (Table S4). Finally, we rewrote the result and discussion sections as suggested to clarify relevant issues.**

1. Line#47-51, there are too few references to represent the minimum and maximum flux rates in lakes, especially given that those references are from lakes in China and Norway only. Also, if such values can be presented with more studies, how would seasonal variations look like in comparison to diurnal ones?

**Answer: We agree with the Reviewer that there are too few studies measuring lake CH$_4$ efflux in the literature and the sampling size and frequency was also different among the limited number of studies (Also see Reviewer #1's comments, specific question 5). Therefore, we deleted the range (maximum and minimum) comparison among lakes and focused on comparing the mean efflux of various lakes in the revised version.**

2. Line#75-78, can you add references for each variable? Line#64-72 well covered the references for each variable, but this section lacks it.

**Answer: We added related references in the method section as suggested in the revised version (Page 4/lines 81-84).**

3. Line#78-82, it sounds like investigating in large lakes is not important. Please rephrase or add some more sentences to justify the importance of this research.

**Answer: We added some sentences and references to emphasize the importance of $CH_4$ emissions from large lakes as suggested (Page 5/lines 88-91).**

4. Line#86-87, I suggest adding references that describe the previous studies, e.g. Liu et al. (2013).

**Answer: Thanks for your suggestion. We added references to describe the previous studies in the revised version (Page 5/ lines 97-102).**

5. Line#109, what are the species names of Carex?

**Answer: The species name of Carex in Poyang Lake is Carex cinerascens Kükenth and Carexargyi Levl.etVant. We added the species scientific names in the revised version (Page 7/ lines 123-124).**

6. Line#128-146, this section can be written more concisely.

**Answer: We rewrote this section as suggested in the revised version.**

7. Line#166, can decreases in $CH_4$ concentrations right after ebullition events be solely explained by diffusion back to lake water? If $CH_4$ molecules were diffused back to the lake water, partial pressure of $CH_4$ inside the chamber should be very high, inhibiting further emission from lake water to chamber. Can they be partially from irregular air mixing inside the chamber, which results in errors in $CH_4$ concentrations? Then, the current method for calculating flux rates needs to be reconsidered.

**Answer: We speculate that the short-term decrease or leveling-off of $CH_4$ concentration inside the chamber after ebullition was mainly caused by the back diffusion of $CH_4$ to surface water due to the high $CH_4$ concentration in the**

**bubbles. This back-diffusion phenomenon has been evidenced for CH₄ efflux over water surfaces (Varadharajan et al., 2010; Wik et al., 2013). The ebullition suddenly increased CH₄ concentration, and thus partial pressure of CH₄, in the chamber headspace, which reversed the normal CH₄ diffusion gradient between surface water and chamber space. We do not think irregular mixing is the main cause in the current study because we had a mixing fan running in each chamber during the whole period of measurement.**

**8.** Line#167-182, this section is confusing. It can be written clearly and concisely.

**Answer: We rewrote this section more clearly and concisely as suggested.**

9. Line#200, were water and sediment samples collected at three sampling points for flux measurements? The paragraph from line#229 can be given in a Table.

**Answer: Yes, we collected water and sediment samples at each of the three sampling sites when taking flux measurements. We added a table (Table S1) to the supplementary material section in the revised version as suggested.**

10. Line#241, T test → t-test

**Answer: Thank you for pointing out the typo. We changed "T test" to t-test as suggested.**

11. Line#242, flux rates are measured three times per season and they may not well represent flux rates of one season of the year. Then, can deviation of these three values be used to quantify inter-annual variability?

**Answer: We agree that 3 measurements in a season for a given year are not enough to represent the seasonal mean CH₄ efflux due to the high temporal variation of the efflux. In the current study, we used 4-year data to compare the seasonal variations, which means 12 data points for each season. We changed the values in Table 1 accordingly by using 12 data points to calculate the seasonal**

**mean effluxes in the revised version (Page 40). For quantifying inter-annual variability we have to interpolate the measured CH$_4$ effluxes to annual efflux through modeling approach. The details of the modeling work were presented in another paper (Liu et al. 2016, in revision). We used the model results to compare the inter-annual, seasonal, and diurnal variabilities of CH$_4$ efflux in the Poyang Lake.**

12. Line#247, please write what b represents in the equation.

**Answer: Thank you for your suggestion. Here b is the exponent of the exponential function between CH$_4$ efflux and sediment temperature. We added it to the text in the revised version (Page 13/lines 275-276).**

13. Line#278, what do you mean by "inconsistent and obvious"?

**Answer: This is a typo. We fixed it in the revised version (Page 15/line 312).**

14. Line#309-331, this part can be written more concisely.

**Answer: Rewritten as suggested.**

15. Line#331-332, sentences of this paragraph do not support this conclusion.

**Answer: We deleted the concluding sentence.**

16. Line#335, here again, can the absolute values be compared with a few references, which are probably based on different observation periods?

**Answer: We agree that comparing the extreme values (minimum and maximum) among different lakes is not much meaningful. So, we deleted the relevant text and focused on comparing diurnal patterns.**

17. Line#338-342, a larger number of data points can produce wider range of values.

**Answer: See answers to question #16.**

**18.** Line#345-356, possible explanations can be added, such as potential drivers that can affect diel CH$_4$ flux patterns and their variations (if measured).

**Answer: Wind speed strongly influenced diel CH$_4$ efflux variations in our study. We discussed this point in the 4.3 section.**

19. 4.2 CH$_4$ effluxes in summer, this section contains a lot of new results, which were not presented in the result section. Also, some sentences describe very detailed information from other studies, which hinders the main focus of the paragraph.

**Answer: We moved them to the "Results" section and rewrote the discussion by focusing on our own results.**

20. 4.3 Timescale dependence of wind, substrate availability, and temperature effects on CH$_4$ effluxes, here again, a lot of new results are reported, such as line#410-414, line#436-451, line#457-461 (repetition from result section), and line#462-468.

**Answer: Again, we moved the results to the "Results" section and rewrote the discussion accordingly.**

21. Line#473-475, considering uncertainties related to infrequent measurements (CH$_4$ efflux rates measured on one day may not represent the mean rates of that month), this kind of statement needs to be corrected.

**Answer: According to our model-based interpolation we found that July had the maximum monthly efflux, while January had the minimum. This conclusion is coincidently in line with the 4-year measurements though we had only 4-day measurements in each month. Therefore, we think that the conclusion still holds.**

22. Table 3, can you add the observation period of each study for better comparison? Also, sorting the rows by lake size and climate would make this Table easier to read.

**Answer: Great idea! We added the observation period of each study and sorted the rows by lake size in the revised version.**

23. Figure 3 and 4, can you add error bars from spatial variability?

**Answer: We added errors bars from spatial variability for Figure 3 and 4 as**

**suggested in the revised version.**

**Reviewer 3**

Specific Comments:

1. Most of the results and discussions were built on the environmental variables and methane flux data. However, there are no data of biogeochemical related environmental variables shown in the figures and tables except Table 2. I would suggest to present the raw data of measured environmental variables in the supplementary material.

**Answer: Thank you for the suggestion. We added a table (Table S1) to the Supplementary Material section to present the raw data of measured environmental variables, such as sediment total nitrogen content, water level, DOC content in the water, and pH in the sediment , in the revised version.**

2. Substrate availability (Line, 432), biological (e.g., microbial activities) and biochemical (e.g., sediment carbon and nitrogen contents processes) (Lines 454-455) are very important factors to link methane efflux to the biogeochemical cycles and understand methane source and sink. Unfortunately, no comprehensive data or evidence to support the role of substrates and microbial activities on methane efflux in this manuscript which could be an important contribution to this journal.

**Answer: We agree with the Reviewer that substrates and microbial activities are important to understanding methane sources and sinks in lakes. In our earlier studies we found that sediment carbon and nitrogen ratio were highly correlated with microbial biomass and community structure (Liu et al., 2015) which was also highly associated with greenhouse gas ($CO_2$, $CH_4$, and $N_2O$) fluxes in the Poyang Lake (Liu and Xu, 2016). In the current study, we focus on examining environmental variables (e.g. climate) that may affect the temporal patterns and**

**variations of CH₄ effluxes in the Poyang Lake. We have added the related information and references to the discussion section in the revised version. Further investigation on the mechanisms of biological and biochemical controls on CH₄ production and oxidation requires lab-based experiments with isotope and microbial DNA sequencing techniques which are beyond the scope of the current study**.

3. It might be a risk to use the data from three sampling sites measured from one day (1 hr? Line 148) to represent methane efflux in that month. For example, it appears a contradiction between high methane efflux measured in July 2011 in Figure 3 and low methane efflux measured in July 2011 in Figure 4a.

**Answer: We measured CH₄ effluxes at monthly interval to examine the seasonal dynamics of the efflux and the value does not necessarily represent the monthly average of CH₄ efflux. We measured CH₄ effluxes from early morning to late afternoon with about 6 cycles of measurements during the day (Pages 10-11/lines 213-219). The values of methane efflux measured in July 2011 in Figure 3 and Figure 4a are different because of different units. CH₄ efflux in Figure 3 was measured on a daily scale, but CH₄ efflux in Figure 4a was based on the hourly scale. So we used different units to present seasonal and diel patterns of CH₄ effluxes.**

4. How long and what time did the authors deploy the floating chambers in the three sampling sites within a day for the study at the large temporal scales (Fig. 3)? I feel 4-year measurements are not a very large temporal scale especially there are no continues measurements/monitering such as deploying floating chamber within a short interval (every week or every two to three days). Since high methane efflux was shown in the early mornings in Fig. 4a, b and d, were the floating chambers deployed

at the same time at three different sites for the data shown in Fig. 3?

**Answer: We measured CH$_4$ fluxes from early morning to late afternoon with about 6 cycles of measurements during the day for the 4-year study. For sampling frequency we measured every monthly. We agree with the Reviewer that 4 year is not "long-term" given the relatively low sampling frequency. So we deleted "long-term" and focused on the multi-seasonal investigations of CH$_4$ effluxes as suggested by Reviewer 1 in the revised manuscript. We used three boats to monitor CH$_4$ fluxes at the three sites, so the floating chambers were deployed at about the same time at the sites as shown in Fig. 3.**

5. The area and water table of Poyang Lake fluctuate dramatically between the wet and dry seasons. The authors only have short but not clear descriptions of the effect of water level on methane efflux, e.g., in Lines 404-405 and Line 432. Methane efflux might be high in dry seasons instead of summer, since methane efflux is expected to be high under lower water level due to decreasing of the hydrostatic pressure (e.g. Chanton et al. 1989). Are there any difference in water level between three sampling sites in different seasons (The mean water depth at three sites should not be always 3m through the whole year; Line 186)? The authors might consider a simple calculation of methane solubility changes due to water level fluctuations to strength the role of water level on methane efflux, e.g., Line 432.

**Answer: It is true that the Poyang Lake features a large seasonal variation of water level, high water level in summer and low in winter. However, the water level at the 3 sites was very similar at a given time of the year. We agree that hydrostatic pressure affects CH$_4$ efflux as reported in Chanton et al. (1989), but our data showed that CH$_4$ efflux was positively correlated with water level. This is because the water level in the Poyang Lake co-varies with other factors,**

**such as temperature and $NH_4^+$ content in the water, which also affect the $CH_4$ efflux throughout the year. For example, we found that the $CH_4$ efflux was highly correlated with sediment temperature at an annual scale. Our results suggest that the $CH_4$ efflux in the Poyang Lake was dominated by temperature rather than water level. The high $CH_4$ efflux in summer was contributed to strong microbial activities induced by warmer temperature and high substrate availability from the flooding water in summer. Therefore, we think the positive correlation between $CH_4$ efflux and water level in the Poyang Lake is a pseudo relation which does not reflect the hydrostatic pressure effect on $CH_4$ efflux as evidenced by Chanton et al. (1989). It is possible to examine the water level effect by calculating $CH_4$ solubility change due to water level fluctuation. However, given the large seasonal variation of temperature in the study area it is very difficult to separate the water level effect based on the $CH_4$ efflux measurements on the water surface. In addition, water level induced $CH_4$ solubility change may affect short-term (minutes to hours) $CH_4$ diffusion gradient and thus $CH_4$ efflux and it should have little impact on $CH_4$ efflux as long as a new diffusion equilibrium has established. Thus, we did not calculate methane solubility changes to further investigate the water level effect on $CH_4$ efflux in revised the manuscript.**

6. As the authors stated in the introduction that methane is driven by three major mechanisms such as molecular diffusion, bubble ebullition and plant-mediated transportation, bubble ebullition is not the only pathway for methane to transport from water to the air. However, data for dissolved methane concentrations in lake water and sediments are lack in this study. No bubble ebullition doesn't mean no methane efflux. I would suggest to include diffusive methane flux to the air for comparison in the

future by analyzing surface water methane concentrations and using the equation from the gas-transfer model e.g., Wanninkhof (1992).

**Answer: Great idea! We will take this suggestion in our future study.**

7. Since many environmental factors and methane fluxes collected in October 2010 in Poyang Lake have been shown in Liu et al., (2013) for spatial studies, the authors may include Liu et al. (2013) in the introduction and discussions to emphasize why the three sampling sites were chosen in this timescale study and the relations between different environmental factors and methane effluxes in Autumn (October).

**Answer: Based on our previous study which examined the spatial variation of CH$_4$ efflux in the Poyang Lake (Liu et al., 2013), we chose the 3 sites which gave CH$_4$ effluxes close to the average efflux of the lake. We provided detailed information of Liu et al. (2013) in the introduction and discussion sections as suggested in the revised version.**

Minor Comments:

1. Lines 57-59: Please add references for the studies in high-latitude, tropical and subtropical lakes.

**Answer: We added references for the studies in high-latitude, tropical and subtropical lakes in the revised version (Page 4/lines 61-63).**

2. Line 129: What fluxes did the floating chamber measured while inserting 20 cm above the water surface?

**Answer: The chamber measured the total CH$_4$ efflux including diffusive and ebullitive fluxes as described in the method section. The plant-mediated CH$_4$ transportation was negligible because no vascular plants grew above water surface at our study sites.**

3. Line 150: the air samples ==> the gas samples

**Answer: Changed as suggested.**

4. Line 159-160; Fig. 4: Since methane efflux was calculated by using a linear regression model to the methane concentration data, should the minimum value be zero instead of a negative value? There should be no negative methane value detected by GC.

**Answer: The negative efflux means CH$_4$-uptake by the lake water due probably to the short-time change in air pressure.**

The magnitude of $CH_4$ emission mainly depends on the dynamic balance between the microbial processes of $CH_4$ production, oxidation, physical transportation from the anaerobic zone to the atmosphere in lakes, and regulation by multiple, interconnected physical, chemical, and biological variables (Sun et al., 2012; Liu et al., 2013; Serrano–Silva et al., 2014; Rasilo et al., 2015). $CH_4$ production and oxidation are microbial processes regulated by organic carbon loading, dissolved organic matter, lake nutrient status, and N availability (Bridgham et al., 2013; Liu et al., 2013; Hershey et al., 2014; Rasilo et al., 2015); temperature (Liikanen et al., 2003; Marotta et al., 2014; Yvon–Durocher et al., 2014); lake depth and size (Juutinen et al., 2009; Rasilo et al., 2015); pH, $O_2$, $NO_3^{2-}$, $Fe^{3+}$, and $SO_4^{2-}$ in the sediment and water column (van Bodegom and Scholten 2001; Schrier–Uijl et al., 2011; Bridgham et al., 2013); and populations and potential activities of methanogens and methanotrophs (Segers, 1998; van Bodegom and Scholten, 2001; Liu et al., 2015, 2016). $CH_4$ transportation is driven by three major mechanisms, namely, molecular diffusion, bubble ebullition, and plant-mediated transportation (Bridgham et al., 2013; Chen et al., 2013; Zhu et al., 2016). These mechanisms are affected by water stratification and seasonal overturns of the water mass, which are determined by temperature (Palma–Silva et al., 2013; Rõõm et al., 2014), wind-forced mixing (Wanninkhof, 1992; Palma–Silva et al., 2013), water depth (Liu et al., 2013), boundary layer dynamics (Poindexter et al., 2015; Anthony and Macintyre, 2016), hydrostatic pressure (Chanton et al., 1989), and

different vascular plants (Juutinen et al., 2009; Zhu et al., 2016). Most studies examined $CH_4$ emissions and their influencing factors in small lakes because of their large contribution to the global $CH_4$ budget (Bastviken et al., 2004; Downing et al., 2010; Bartosiewics et al., 2015; Holgerson et al., 2016). Although small lakes are a large source of atmospheric $CH_4$, $CH_4$ emissions from large lakes was not neglected due to their large areas (Bastviken et al., 2010; Rasilo et al., 2015; Townsend-Small et al., 2016). However, few studies reported temporal $CH_4$ emissions and their key regulating factors at different temporal scales in large lakes. Therefore, investigating the impacts of physical and biological factors on temporal $CH_4$ effluxes based on multi-seasonal long-term measurements in a large lake is also important to estimate lake $CH_4$ emissions.

Poyang Lake, a subtropical lake, is the largest freshwater lake in China, but its annual multi-seasonal $CH_4$ emissions have not been adequately measured. In our previous study, we have explored the spatial variations of $CH_4$ efflux over the lake with 44 sampling locations (Liu et al., 2013). In addition, we also found that microbial biomass and community structure highly influenced $CH_4$ efflux in Poyang Lake (Liu et al., 2016). In this study, we measured the $CH_4$ efflux in three sites which we chose on the basis of our previous result over the course of 4 years in Poyang Lake to (1) examine the annual multi-seasonal mean $CH_4$ efflux; (2) explore the $CH_4$ efflux dynamics, including diel, and seasonal, and inter annual variations; and (3) quantify the relationships between the $CH_4$ efflux and environmental factors, and identify the possible factors driving $CH_4$ effluxes at different temporal scales.

**2. Materials and methods**

**2.1. Site description**

Poyang Lake (28°22′–29°45′N, 115°47′–116°45′E) is located in Southern China in Jiangxi Province, with a surface area of 3283 km$^2$ and a total catchment area of 162,000 km$^2$, which is separated to the northern and southern parts by the Songmen Mountain. Poyang Lake receives water input from five main tributaries, namely, the Raohe River, Xinjiang River, Fuhe River, Ganjiang River, and Xiushui River. The climate is humid subtropical with a mean annual temperature of 17.5 °C and an annual precipitation of 1680 mm (Ye et al., 2011). Vegetation in the lake is composed of macrophytes, including *Carex* sp. (Carex cinerascens Kükenth and Carexargyi Levl.etVant) and *Artemisia selengensis* in the hydrophyte zone, and the main submerged aquatic macrophytes, including *Ceratophyllum demersum*, *Potamogeton malaianus*, *Potamogeton crispus*, and *Hydrilla verticillata* (Wang et al., 2011).

This study was conducted near the Poyang Lake Laboratory of the Wetland Ecosystem Research Station (operated by the Chinese Academy of Sciences), which is located in the northern sub-basin of Poyang Lake in Xingzi County, Jiangxi Province (Fig. 1). The five tributaries flow into the lake in the southeast of Xingzi County, which then joins with the Yangtze River. The water level fluctuated dramatically from 7.78 m to 18.57 m above sea level (Wu Song) between the wet (April to September) and dry seasons (October to March) during the study period because of rainfall and Three Gorge management. Poyang Lake is not stratified (Zhu and Zhang, 1997), with mean and maximum depths of 8 and 23 m, respectively. The mean concentrations of total nitrogen (TN), total phosphorous (TP), suspended solids (SS), and chlorophyll *a* (Chl *a*) in the Poyang lake were 3.45, 0.11, 39.98, and 9.04 mg L$^{-1}$, respectively (Yao et al., 2015).

**2.2. CH$_4$ efflux measurements**

The CH$_4$ efflux was measured using floating chambers, including both ebullition and diffusive fluxes (Bastviken et al., 2004, 2010). The floating chamber was fabricated using a PVC pipe 100 cm in length and 20 cm in diameter with Styrofoam floats attached to the sides. The floating chambers were inserted 80 cm into the water and 20 cm above the water surface to minimize the perturbation of the surface water flow to the pressure inside the chambers. We tested the chamber system with different insertion depths in the laboratory and field, and found that the current depth of about 80 cm could effectively prevent the impacts of the surrounding Styrofoam floats while maintaining the chamber balance in moderate winds. A similar design of floating chambers was used in previous studies (Lorke et al., 2015; Zhao et al., 2015). Zhao et al. (2015) have recently conducted a systematic comparison of the effects of chamber shape, dimension, and insertion depth into the water on CH$_4$ effluxes and found that insertion depth only slightly affects the CH$_4$ efflux measured in the Three Gorges Reservoir when wind speed is relatively low. In the current study, the insertion depth was deeper than those of previous studies to avoid the impact of waves in Poyang Lake on the chamber body. Earlier studies also found that floating chambers should be seated at the water surface with minimal insertion into the water in a flowing-water system to minimize the "drag" effect of flowing water on chamber pressure (Bastviken et al., 2010; Vachon et al., 2013; McGinnis et al., 2015). Except for some waves, However, the water in Poyang Lake did not have an apparent directional flow except for some waves during the measurement period. So In the current study, the insertion depth was deeper than those of previous studies to avoid the impact of waves in Poyang Lake on the chamber body in the current study. A detailed description of the floating chamber system can be found in Liu et al. (2013). So we measured the

150    total CH$_4$ efflux including both ebullition and diffusive effluxes and cannot

differentiate ebullitive and diffusive fluxes by our chamber.

[revised manuscript text omitted]

effects models in order to take into account CH$_4$ efflux variations among three sites

when we investigated seasonal and diurnal variations as well as the relationships

260    between CH$_4$ efflux and environmental variables. We used the Vant′ Hoff equation to

calculate the temperature sensitivity ($Q_{10} = e^{10b}$, where b is the exponent of the

exponential function between CH$_4$ efflux and sediment temperature) of CH$_4$ efflux

(Xu and Qi, 2001; Wei et al., 2015). All statistical analyses were performed using the

SPSS 17.0 statistical software (SPSS Inc., Chicago, IL, USA), and graphs were

265    created using the Sigma Plot 11.0 program (Systat Software Inc., San Jose, CA,

USA).

**3. Results**

3.1. $CH_4$ effluxes in Poyang Lake

~~The mean $CH_4$ efflux was 0.54 ± 0.053 mmol m$^{-2}$ day$^{-1}$ in Poyang Lake over the 4-year period, with annual mean effluxes of 0.47 ± 0.54, 0.56 ± 0.41, 0.52 ± 0.55, and 0.60 ± 0.56 mmol m$^{-2}$ day$^{-1}$ in 2011, 2012, 2013, and 2014, respectively (Table 1). The inter-annual variation of $CH_4$ efflux was moderately high with a CV of 9.8% over the 4 years. The mean $CH_4$ efflux in 2014 was 25.7% greater than that in 2011, justifying the necessity for long-term measurements.~~

3.1.1. Seasonal $CH_4$ effluxes

The seasonal variations of $CH_4$ effluxes in Poyang Lake were prominent, demonstrating a similar pattern to that of seasonal temperature (Fig. 3). In general, the annual maximum $CH_4$ effluxes occurred in summers and the minimum in winters. The $CH_4$ efflux increased slowly in early spring and then rapidly in May, reaching its maximum in July. After reaching the maximum, the $CH_4$ efflux decreased sharply in August and September and then slowly before reaching its minimum in January (Fig. 3). Significant differences in the mean $CH_4$ effluxes existed between summers and the other three seasons throughout the 4 years ($p < 0.05$), whereas the differences in the $CH_4$ effluxes among the spring, autumn, and winter seasons were not statistically significant ($p > 0.05$) (Table 1). Additionally, the site effect was not statistically significant over the 4-year period (Table S2). The differences among the three sites were minor with the 4-year mean of 0.53 mmol m$^{-2}$ day$^{-1}$, 0.55 mmol m$^{-2}$ day$^{-1}$, and 0.54 mmol m$^{-2}$ day$^{-1}$ respectively. In particular, the seasonal patterns of $CH_4$ effluxes at the three sites were similar and also in line with the seasonal pattern averaged over the three sites.

3.1.2. Diel CH$_4$ effluxes

The CH$_4$ effluxes in Poyang Lake also exhibited apparent variations within a day because the daily maximum appeared late in the morning (10:00–12:00 h) and the minimum early in the morning the next day (4:00–6:00 h). The diel pattern of the CH$_4$ efflux was asymmetric, fast increasing in the morning from 8:00 h to 12:00 h and slowly decreasing in the afternoon and during the night, especially in the summer (Fig. 4). However, the diel pattern of the CH$_4$ efflux was inconsistent . For example, the diel pattern on January 13–14, 2013 was an exception, when the maximum efflux occurred around 6:00 h on January 14$^{th}$ and a severe cold front with heavy fogs enveloped the Poyang Lake area in the early morning of January 14$^{th}$. The CH$_4$ efflux magnitudes were significantly larger during summer compared to winter.  The CH$_4$ efflux could also change abruptly throughout a day. For example, the efflux sharply dropped from 0.068 to −0.012 mmol m$^{-2}$ h$^{-1}$ within barely 2 h, as observed on July 23, 2011, indicating that the lake switched from a CH$_4$ source to sink within a short period of time (Fig. 4a). This abrupt change was also observed in the afternoon of August 28, 2012 (Fig. 4b). Furthermore, we compared the differences of diurnal patterns at each sites for the four diel sampling. Our results showed that the diel patterns of CH$_4$ effluxes were similar in the three sites for each diel investigation (Fig. 5) and the site effect was not statistically significant (Table S3). The diel pattern of the CH$_4$ efflux was somewhat different during certain hours such as from 22:00 h to 00:00h on July 24$^{th}$ in 2011.

3.2. Relationships between CH$_4$ efflux and environmental variables

**3.2.1 Simple regression relationships between $CH_4$ efflux and environmental variables**

In our study, $CH_4$ effluxes increased exponentially with sediment temperature for both in the summer and in other seasons (Fig. 6). The $CH_4$ effluxes were more sensitive to temperature in the summers than in other seasons. The temperature sensitivity, indicated by the $Q_{10}$ values, was 2.04 and 1.67 in the summer and other three seasons, respectively (Fig. 6).

We found that $CH_4$ effluxes were also highly associated with other climate and environmental variables in both lake water and sediments. We found that $CH_4$ effluxes were negatively correlated with $NH_4^+$, TN and DO concentrations in the lake water, but positively with Chl $a$ content in the water and TN content in the sediment (Table 2). Furthermore, we found that other environmental factors, such as DOC content in the water, pH in the water and in the sediment, $NO_3^-$ concentrations in the water and in the sediment, COD, and TP in the water, had insignificant ($p > 0.05$) relationships with $CH_4$ effluxes in Poyang Lake.

In the current study, we also found that the relationship between $CH_4$ effluxes and wind speed was scale-dependent. At the diel scale, wind speed was significantly correlated ($p < 0.03$) with $CH_4$ effluxes for the average of the 3 sites at the diel scale (Fig. 7), but was weakly correlated with $CH_4$ effluxes at the diurnal and seasonal scales (Fig. 7). In addition, the relationships between wind speed and $CH_4$ effluxes for each individual site were similar with the relationships for the average of the 3 sites though the regression coefficients for each individual site were slightly different, but not statistically significant ($p > 0.25$).

**3.2.2 Multiple regression relationships between $CH_4$ efflux and environmental variables**

In the current study, environmental factors differed in importance depending on

the timescale in the stepwise multiple regressions analyses. The results of stepwise multiple regressions on a seasonal scale showed that the sediment temperature, sediment TN content, DO, and TP content in the water were significant predictors of CH$_4$ effluxes (Table 23). It should be noted that multicollinearity didn't occurred among these significant variables (Table S4). In specific, sediment temperature and sediment TN content explained 65% of the variation in CH$_4$ effluxes for 4 years when we used the first group of factors. The sediment temperature and TN content explained 73% of the CH$_4$ efflux variations when the second group of variables was added to the first group. The sediment temperature, sediment TN content, DO, and TP contents in the water explained 89% of the CH$_4$ efflux variation when the three groups of variables were used together. Wind speed was the only significant variable for the CH$_4$ efflux variations on a diel scale. Wind speed explained 58%, 56%, 84% and 86% of the CH$_4$ efflux variations in 24–25 July 2011, 5–6 September 2012, 13–14 January 2013 and 14–15 January 2015, respectively (Figs. 5a7a-5d7d). In addition, the same environmental variables were selected in the final model for each individual site as for the average of the 3 sites though the regression coefficients were slightly different (Table S5), but not statistically significant (p > 0.20).

[revised manuscript text omitted]

van Bodegom, P. M., Scholten, J. C. M.: Microbial processes of $CH_4$ production in a rice paddy soil: Model and experimental validation, Geochimica et Cosmochimica Acta, 65, 2055-2066, 2001.

790   Verpoorter, C., Kutser, T., Seekell, D. A., Tranvik, L. J.: A global inventory of lakes based on high-resolution satellite imagery, Geophys. Res. Lett., 41, 6396-6402, 2014.

Wang, Y. Y., Yu, X. B., Li, W. H., Xu, J., Chen, Y. W., Fan, N.: Potential influence of water level changes on energy flows in a lake food web, Chin. Sci. Bull., 56,
795   2794-2802, 2011.

Wanninkhof, R.: Relationship between wind speed and gas exchange over the ocean, J. Geophys. Res., 97, 7373-7382, 1992.

Wei, D., Xu, R., Tarchen, T., Wang, Y. S., Wang, Y. H.: Considerable methane uptake by alpine grasslands despite the cold climate: *in situ* measurements on the central Tibetan Plateau, 2008-2013, Glob. Change Biol., 21, 777-788, 2015.

Wik, M., Thornton, B. F., Bastviken, D., Maclntyre, S., Varner, R. K., Crill, P. M.: Energy input is primary controller of methane bubbling in subarctic lakes, Geophys. Res. Lett., 41, 555-560, 2014.

Wetzel, R.G.: Limnology: Lake and River ecosystems, In: Academic Press. A Harcourt Science and Technology Company, pp: 24570-24577, 2001.

Xiao, S. B., Liu, W. G., Yang, H., Liu, D., Wang, Y., Peng, F.: Extreme methane bubbling emissions from a subtropical shallow eutrophic pond, Austin Biometrics and Biostatistics, 1, 1-6, 2015.

Xiao, S. B., Wang, Y. C., Liu, D. F., Yang, Z. J., Lei, D., Zhang, C.: Diel and seasonal variation of methane and carbon dioxide fluxes at Site Guojiaba, the Three Gorges Reservoir, J. Environ. Sci., 25, 2065-2071, 2013.

Xing, Y. P., Xie, P., Yang, H., Ni, L.Y., Wang, Y. S., Tang, W. H.: Diel variation of methane fluxes in summer in a eutrophic subtropical Lake in China, J. Freshw. Ecol., 19, 639-644, 2004.

Xing, Y. P., Xie, P., Yang, H., Ni, L.Y., Wang, Y. S., Rong, K.W.: Methane and carbon dioxide fluxes from a shallow hypereutrophic subtropical lake in China, Atmos. Environ., 39, 5532-5540, 2005.

Xing, Y. P., Xie, P., Yang, H., Wu, A. P., Ni, L.Y.: The change of gaseous carbon fluxes following the switch of dominant producers from macrophytes to algae in a shallow subtropical lake of China, Atmos. Environ., 40, 8034-8043 2006,.

Xu, M., Qi, Y.: Spatial and seasonal variations of $Q_{10}$ determined by soil respiration measurements at a Sierra Nevadan forest, Glob. Biogeochem. Cy., 15, 687-696, 2001.

Yang, H., Xie, P., Ni, L.Y., Flower, R. J.: Underestimation of $CH_4$ Emission from

825        Freshwater Lakes in China. Environ, Sci. Technol., 45, 4203-4204, 2011.

Yang, S. S., Chen, I. C., Liu, C. P., Liu, L.Y., Chang, C. H.: Carbon dioxide and methane emissions from Tanswei River in Northern Taiwan, Atmos. Pollut. Res., 6, 52-61, 2015.

Yao, X., Wang, S. R., Ni, Z. K., Jiao, L. X.: The response of water quality variation in

830        Poyang Lake (Jiangxi, People's Republic of China) to hydrological changes using historical data and DOM fluorescence, Environ. Sci. Pollut. Res., 22, 3032-3042, 2015.

Ye, X. C., Zhang, Q., Bai, L., Hu, Q.: A modeling study of catchment discharge to Poyang Lake under future climate in China, Quatern. Int., 244, 221-229, 2011.

835      Yvon-Durocher, G., Allen, A. P., Bastviken, D., Conrad, R., Gudasz, C., St-Pierre, A., Thanh-Duc, N., del Giorgio, P. A.: Methane fluxes show consistent temperature dependence across microbial to ecosystem scales, Nature, 507, 488-491, 2014.

Zhao, Y., Sherman, B., Ford, P., Demarty, M., DelSontro, T., Harby, A., Tremblay, A., Øverjordet, I.B., Zhao, X. F., Hansen, B. H., Wu, B.: A comparison of methods

840        for the measurement of $CO_2$ and $CH_4$ emissions from surface water reservoirs: Results from an international workshop held at Three Gorges Dam, June 2012, Limnol. Oceanogr.: Methods, 13, 15-29, 2015.

Zhu, H. H., Zhang, B.: Poyang Lake-hydrology、biology、sediment、wetland、exploitation andrenovation, In: University of science and technology of China

845        Press. Hefei, pp. 97-99, 1997.

Zhu, D., Wu, Y., Chen, H., He, Y. X., Wu, N.: Intense methane ebullition from open water area of a shallow peatland lake on the eastern Tibetan Plateau, Sci. Total Environ., 542, 57-64, 2016.

850

Table 1 Seasonal  mean of $CH_4$ effluxes with the chamber measurements in Poyang Lake

|  |  |  |  |  |
|---|---|---|---|---|
|  |  |  |  |  |
|  |  |  |  |  |
|  |  |  |  |  |
|  |  |  |  |  |
|  |  |  |  |  |

| Season | $CH_4$ efflux (mmol m$^{-2}$ day$^{-1}$) |
|---|---|
| Spring (Mar–May) | 0.30±0.11bd |
| Summer (Jun–Aug) | 1.34±0.32a |
| Autumn (Sep–Nov) | 0.33±0.14b |
| Winter (Dec–Feb) | 0.18±0.077cd |

**Note**: Means with different letters are significantly different as determined by multiple comparisons on a seasonal scale (one-way ANOVA, post hoc Tukey test, $p < 0.05$) .

Table 2 Correlation relationship between seasonal $CH_4$ efflux and environmental factors

| Environmental factors | Correlation coefficient | Environmental factors | Correlation coefficient |
|---|---|---|---|
| Dissolved oxygen | -0.74** | Sediment- $NO_3^-$ | -0.2 |
| Sediment nitrogen | 0.37* | Sediment-pH | -0.13 |
| Sediment carbon | 0.24 | Water-COD | -0.016 |
| pH in the water | -0.29 | Water-$NO_3^-$ | -0.24 |
| Sediment C/N | -0.064 | Water-$NH_4^+$ | -0.36* |
| Conductivity | -0.37 | Water-chla | 0.46* |
| Wind speed | 0.008 | Water-TN | -0.35* |
| DOC | -0.015 | Water-TP | 0.11 |

**Note**: Asterisks indicate statistically significant differences between $CH_4$ efflux and environmental factors (one asterisk, $p < 0.05$; two asterisks, $p < 0.01$).

Table 2-3 Multivariate regressions between seasonal CH$_4$ efflux and environment factors

| No. | Number of variables | Regression Equation | n | R$^2$ | p |
|---|---|---|---|---|---|
| Group 1 | 12 | EffluxCH$_4$ = −10.48 + 110.57 ST + 65.06SNC | 48 | 0.65 | 0.004 |
| Group1 + Group 2 | 16 | EffluxCH$_4$ = −12.66 + 0.57ST + 90.81SNC | 43 | 0.73 | 0 |
| Group 1 + Group 2 + Group 3 | 19 | EffluxCH$_4$ = −3.89 + 0.56ST + 102.88SNC − 35.56TP − 0.74DO | 19 | 0.89 | 0 |

Note: Nd means that no variable input to the stepwise regression exists. Variables in group 1 included sediment temperature (ST), sediment total nitrogen content (SNC), water level, DOC content in the water, pH in the sediment, NH$_4^+$ and NO$_3^-$ concentrations in the water and in the sediment, sediment organic carbon content, the ratio of carbon and nitrogen, and the mean daily wind speed. Variables in group 2 included TN, TP, COD, and Chl *a* contents in the water. Variables in group 3 included DO content, conductivity, and pH in the water.

870

Table 3 4 Mean CH$_4$ effluxes in Poyang Lake in comparison with other large lakes

| Lake | Lake size (km²) | Region | Climate | CH$_4$ efflux (mmol m$^{-2}$ day$^{-1}$) | References |
|---|---|---|---|---|---|
| 11 lakes | 1 | Laurentians, Canada | Boreal | 4.08 | Rasilo et al., 2015 |
| 26 lakes | 47 | Chicoutimi, Canada | Boreal | 1.08 | Rasilo et al., 2015 |
| 27 lakes | 41 | Abitibi, Canada | Boreal | 1.67 | Rasilo et al., 2015 |
| 16 lakes | 171 | Chibougamau, Canada | Boreal | 0.17 | Rasilo et al., 2015 |
| 20 lakes | 7 | James Bay, Canada | Boreal | 1.08 | Rasilo et al., 2015 |
| 45 lakes | 5 | Côte-Nord, Canada | Boreal | 1.17 | Rasilo et al., 2015 |
| 31 lakes | 2 | Eastmain, Canada | Boreal | 0.58 | Rasilo et al., 2015 |
| 48 lakes | 242 | Scheffervill, Canada | Boreal | 0.42 | Rasilo et al., 2015 |
| Lake Mendota | 39.4 | North America | Boreal | 0.50 | Fallon et al.,1980 |
| Dillon | 13 | North America | Boreal | 0.61 | Smith and Lewis,1992 |
| Fiolen | 1.5 | Sweden | Boreal | 0.02 | Bastviken et al., 2004 |
| Kevätön | 4 | Finland | Boreal | 0.22 | Huttunen et al., 2003 |
| Biwa | 674 | Japan | Temperate | 0.27 | Miyajima et al,1997 |
| Kasumigaura | 168 | Japan | Temperate | 0.26 | Utsuumi et al.,1998a |
| Nojiri | 4.4 | Japan | Temperate | 0.06 | Utsuumi et al.,1998b |
| 5 lakes | range 1-11, 3436[a] | Netherlands | Temperate | 5.85 | Schrier-Uijl et al., 2011 |
| Donghu | 27.9 | China | Subtropical | 1.46 | Xing et al., 2005 |
| TR lake | 71.4 | Pantanal, South America | Tropical | 0.65[b]/5.74[c] | Bastviken et al., 2010 |
| BB lake | 36.3 | Pantanal, South America | Tropical | 0.50[b]/5.63[c] | Bastviken et al.,2010 |
| Biandantang | 3.3 | China | Subtropical | 1.32 | Xing et al., 2006 |
| Võrtsjärv | 270 | Estonia | Boreal | 1.28[b]/2.09[c] | Rõõm et al., 2014 |
| 43 lakes | range1-10, 782073.8[a] | worldwide | Mainly boreal | 0.12 | Holgerson and Raymond, 2016 |
| 18 lakes | range10-100, 597789.3[a] | worldwide | Mainly boreal | 0.10 | Holgerson and Raymond, 2016 |
| 6 lake | >100, 2024015.8[a] | worldwide | Mainly boreal | 0.06 | Holgerson and Raymond, 2016 |
| Poyang Lake | 3283 | China | Subtropical | 0.54 | Present study |

875

| Lake | Lake size (km$^2$) | Region | Climate | CH$_4$ efflux (mmol m$^{-2}$ day$^{-1}$) | References | Sampling period |
|---|---|---|---|---|---|---|
| 11 lakes | 1 | Laurentians, Canada | Boreal | 4.08 | Rasilo et al., 2015 | 11/0[D] |
| Fiolen | 1.5 | Sweden | Boreal | 0.02 | Bastviken et al., 2004 | Once |
| 31 lakes | 2 | Eastmain, Canada | Boreal | 0.58 | Rasilo et al., 2015 | 14/17*[D] |
| Kevätön | 4 | Finland | Boreal | 0.22 | Huttunen et al., 2003 | 12 times |
| 45 lakes | 5 | Côte-Nord, Canada | Boreal | 1.17 | Rasilo et al., 2015 | 45/0[D] |
| 20 lakes | 7 | James Bay, Canada | Boreal | 1.08 | Rasilo et al., 2015 | 14/6[D] |
| Dillon | 13 | North America | Boreal | 0.61 | Smith and Lewis,1992 | 9 times |
| Lake Mendota | 39.4 | North America | Boreal | 0.5 | Fallon et al.,1980 | 6 times |
| 27 lakes | 41 | Abitibi, Canada | Boreal | 1.67 | Rasilo et al., 2015 | 21/6[D] |
| 26 lakes | 47 | Chicoutimi, Canada | Boreal | 1.08 | Rasilo et al., 2015 | 19/7[D] |
| 16 lakes | 171 | Chibougamau, Canada | Boreal | 0.17 | Rasilo et al., 2015 | 14/2[D] |
| 48 lakes | 242 | Scheffervill, Canada | Boreal | 0.42 | Rasilo et al., 2015 | 48/0[D] |
| Võrtsjärv | 270 | Estonia | Boreal | 1.28[B]/2.09[C] | Rõõm et al., 2014 | 21 times |
| 6 lake | >100, 2024015.8[A] | worldwide | Mainly boreal | 0.06 | Holgerson and Raymond, 2016 | Multiple times |
| 18 lakes | range10-100, 597789.3[A] | worldwide | Mainly boreal | 0.1 | Holgerson and Raymond, 2016 | Multiple times |
| 43 lakes | range1–10, 782073.8[A] | worldwide | Mainly boreal | 0.12 | Holgerson and Raymond, 2016 | Multiple times |
| Nojiri | 4.4 | Japan | Temperate | 0.06 | Utsuumi et al.,1998b | 6 times |
| 5 lakes | range 1–11, 3436[A] | Netherlands | Temperate | 5.85 | Schrier–Uijl et al., 2011 | twice |
| Kasumigaura | 168 | Japan | Temperate | 0.26 | Utsuumi et al.,1998a | 72 times |
| Biwa | 674 | Japan | Temperate | 0.27 | Miyajima et al.,1997 | 3 times |
| Biandantang | 3.3 | China | Subtropical | 1.32 | Xing et al., 2006 | 12 times |
| Donghu | 27.9 | China | Subtropical | 1.46 | Xing et al., 2005 | 48 times |
| Poyang Lake | 3283 | China | Subtropical | 0.54 | Present study | 48 times |
| BB lake | 36.3 | Pantanal, South America | Tropical | 0.50[B]/5.63[C] | Bastviken et al.,2010 | Once |
| TR lake | 71.4 | Pantanal, South America | Tropical | 0.65[B]/5.74[C] | Bastviken et al., 2010 | Once |

Note: A means total areas in the given lake size. B means diffusive effluxes and C means total effluxes, including diffusion and ebullition. D means number of lakes measured once/twice. * means 24/0 in 2006, 8/11 in 2007, 0/13 in 2008, 2/10 in 2009, respectively.

**Figure Captions**

Figure 1. Location of sampling sites in Poyang Lake.

Figure 2. Examples of calculating the slope of total effluxes, including diffusive and ebullitive effluxes. All the concentrations are presented in original (volumetric parts per million-units). White circles represent the $CH_4$ concentrations at different sampling times. Grey circles represent the adjusted concentration. Black trendlines represent the data used for the total efflux calculation. The different letters in the figure panels mean different occurrence times for ebulltion: no ebullition (a), occurrence of ebullition at 20 min (b), 40 min (c), and 60 min (d), respectively.

Figure 3. Seasonal variations of $CH_4$ effluxes and sediment temperatures in Poyang Lake. White circles represent the variation of $CH_4$ effluxes, and black circles describe the variation of sediment temperature in the 4-year period.

Figure 4. Diel variations of $CH_4$ effluxes in Poyang Lake. Different panels present the diel variations of the $CH_4$ effluxes in 24–25 July 2011 (a), 5–6 September 2012 (b), 13–14 January 2013 (c), and 14–15 January 2015 (d). White circles describe the diel variations of the $CH_4$ effluxes. Horizontal short dashed lines mean the average value of the diel $CH_4$ effluxes.

Figure 5. Diel variations of $CH_4$ effluxes among three sites.

 Different panels present the diel variations of the $CH_4$ effluxes in 24–25 July 2011 (a), 5–6 September 2012 (b), 13–14 January 2013 (c), and 14–15 January 2015 (d).

Relationships between $CH_4$ effluxes and wind speed in Poyang Lake.

White circles represent the observed values of $CH_4$ effluxes and wind speed. Different panels mean the variations of $CH_4$ effluxes at a bihourly interval within a day, including in 24–25 July 2011 (a), 5–6 September 2012 (b), 13–14 January 2013 (c), and 14–15 January 2015 (d), on a diurnal scale (e), and on a seasonal scale (f). Panels e and f include all the measurements during the observation period. We excluded the white-crossed circle in figure e in the regression analysis because of a severe cold front.

Figure 6. Relationship between sediment temperature and $CH_4$ effluxes in Poyang Lake.

White circles represent the observed values of the diurnal mean $CH_4$ effluxes and sediment temperature in summer, and black circles represent the observed values of the diurnal mean $CH_4$ effluxes and sediment temperature in the other seasons in the 4-year period. Black lines represent the fitting curves of the relationship between $CH_4$ effluxes and sediment temperature.

Figure 7. Relationships between $CH_4$ effluxes and wind speed in Poyang Lake.

White circles represent the observed values of $CH_4$ effluxes and wind speed. Different panels mean the variations of $CH_4$ effluxes at a bihourly interval within a day, including in 24–25 July 2011 (a), 5–6 September 2012 (b), 13–14 January 2013 (c), and 14–15 January 2015 (d), on a diurnal scale (e), and on a seasonal scale (f). Panels e and f include all the measurements during the observation period. We excluded the white-crossed circle in figure c in the regression analysis because of a severe cold front.

925

Fig. 1

[Figure]

[Figure]

Fig. 2

[Figure]

955

960

965

970

Fig. 3

[Figure]

悬挂缩进：2 字符

975

980

985

990

[Figure]
Fig. 4

[Figure]

悬挂缩进：2 字符

[Figure]

[Figure]

Fig.5

[Figure]

[Figure]

悬挂缩进：2 字符

1015

[Figure]

1020

1025

1030